# Beyond Imitation: Leveraging Fine-grained Quality Signals for Alignment

**Geyang Guo**[1]*, **Ranchi Zhao**[1]*, **Tianyi Tang**[1], **Wayne Xin Zhao**[1]†, **Ji-Rong Wen**[1,2]

[1]Gaoling School of Artificial Intelligence, Renmin University of China.
[2]School of Information, Renmin University of China.
`guogeyang@ruc.edu.cn, ranchizhao@gmail.com,`
`steventianyitang@outlook.com, batmanfly@gmail.com, jrwen@ruc.edu.cn`

## Abstract

Alignment with human preference is a desired property of large language models (LLMs). Currently, the main alignment approach is based on reinforcement learning from human feedback (RLHF). Despite the effectiveness of RLHF, it is intricate to implement and train, thus recent studies explore how to develop alternative alignment approaches based on supervised fine-tuning (SFT). A major limitation of SFT is that it essentially does imitation learning, which cannot fully understand what are the expected behaviors. To address this issue, we propose an improved alignment approach named **FIGA**. Different from prior methods, we incorporate fine-grained (*i.e.,* token or phrase level) quality signals that are derived by contrasting good and bad responses. Our approach has made two major contributions. Firstly, we curate a refined alignment dataset that pairs initial responses and the corresponding revised ones. Secondly, we devise a new loss function can leverage fine-grained quality signals to instruct the learning of LLMs for alignment. Extensive experiments have demonstrated the effectiveness of our approaches by comparing a number of competitive baselines. We release all the above-mentioned resources at https://github.com/RUCAIBox/FIGA.

## 1 Introduction

Pre-trained large language models (LLMs) such as LLaMA (Touvron et al., 2023a) have shown remarkable potentials to solve various downstream tasks by mastering the universal pre-training task of next-token prediction. While after large-scale pre-training, it often needs subsequent tuning for enhancing and regulating the behaviors of LLMs. Two typical approaches are supervised fine-tuning (SFT) and reinforcement learning from human feedback (RLHF), which can largely improve LLMs in both task solving capacity and human alignment (Ouyang et al., 2022).

Despite widely explored, SFT and RLHF have their own strengths and weaknesses. On the one hand, SFT is easy to implement and can effectively boost the general task solving abilities by instruction based eliciting (Wei et al., 2021; Ouyang et al., 2022; Chung et al., 2022), while it mainly imitates the behaviors of experts (essentially doing *behavior clone* (Wiseman & Rush, 2016)), which are demonstrated by the human annotators or powerful LLMs such as ChatGPT. Therefore, the SFT performance highly relies on high-quality demonstration data (Zhou et al., 2023), and might suffer from the huge distribution shifts between its outputs and imitated outputs (Zhang et al., 2019; Schulman, 2023; Zhao et al., 2023a). On the other hand, RLHF can better explore the semantic space of LLMs, and identify the optimal policy by encouraging good behaviors and discouraging bad behaviors during learning. However, it is very complicated to effectively implement, often suffering from training instability issues such as reward collapse (Song et al., 2023; Wolf et al., 2023).

To leverage the benefits of SFT and RLHF, several recent studies propose to develop alignment approaches without reinforcement learning (RL). These studies typically construct refined instruction data using methods such as quantile ranking (Lu et al., 2022) and rejection-sampling (Touvron et al.,

---

*Equal contribution.
†Corresponding author.

2023b), and then follow or slightly modify the original SFT loss. Another line of research designs alternative optimization approaches that bypasses reward modeling (Rafailov et al., 2023). To conduct effective alignment without RL, a key issue is how to effectively learn by discriminating good and bad behaviors as that in RLHF (Ouyang et al., 2022), such that LLMs can understand what are good behaviors to follow and what are bad behaviors to avoid. Despite the prior efforts, they are largely limited by response-level discrimination signals: they are only aware of the quality label (*e.g.,* good or bad) of a demonstration but not what makes it good or bad. Thus, it can't fully capture the correct alignment behaviors even demonstrated by what are good and bad behaviors.

In this work, we introduce **FIGA**, a novel method that aligns language models with human preferences. The core idea is to contrast a low-quality initial response from a LLM's output with a corresponding high-quality revised response by another powerful LLM (*e.g.,* ChatGPT), so that LLMs can be noted with *what are newly added* (good actions) and *what are removed or substituted* (bad actions) from such a revision process. Such fine-grained quality signals can be more useful than the widely used response-level quality signal. It can instruct LLMs to emphasize the learning of good actions and penalize the bad actions in a single response. To implement our approach, we first curate an alignment dataset called *SPA* that pairs an initial response with a revised response under the guidance of the ground-truth demonstrations. We mainly keep the queries that a LLM performs less well on, and perform strict filtering. Further, we design a new fine-tuning method that assigns specific token-level weights to different parts (*e.g.,* good or bad tokens). Our learning loss can directly impose fine-grained reward scores to guide the learning of LLMs for improved alignment.

To the best of our knowledge, it is the first attempt that leverages fine-grained quality signals for improving the alignment of LLMs without RL. Our approach can make LLMs better understand what are good and bad behaviors beyond simple imitation. By conducting extensive experiments, we demonstrate that **FIGA** shows promising performance in aligning language models with human preferences: our approach outperform the initial supervised-finetuned model by notable 3.2 points and the strong PPO method by 1.8 points.

## 2 RELATED WORK

In this section, we review the related work in the two aspects, namely reinforcement learning from human feedback and alignment without reinforcement learning.

**Reinforcement learning from human feedback**   Large-scale pre-training empowers large language models (LLMs) to acquire extensive knowledge, underscoring their remarkable potential across diverse tasks (Brown et al., 2020; Kojima et al., 2022; Zhang et al., 2022; Chowdhery et al., 2022). Nonetheless, models exclusively focus on next token prediction in pre-training phrase, while do not consider human preferences. Consequently, this gives rise to unexpected behaviors like harmful or inaccurate information, and emphasizes the necessity to align language models with human preferences. The current mainstream approaches (Ouyang et al., 2022) to better harness the capabilities of LLMs include supervised fine-tuning (SFT) and reinforcement learning from human feedback (RLHF). To be specific, this involves three stages: firstly, using SFT to enable the model to better follow human instructions; subsequently, training a reward model (RM) using human preference data; and ultimately, tune the model to maximize the reward through the proximal policy optimization (PPO) (Schulman et al., 2017) algorithm. Furthermore, there are works exploring enhancement for this process (Ramamurthy et al., 2022; Lightman et al., 2023; Lee et al., 2023). However, RLHF presents challenges due to complex coding and hyper-parameters selecting. Besides, it requires loading three to four models simultaneously, resulting in high memory usage. These challenges propel researchers to explore alternative approaches to align language models with human feedback.

**Alignment without reinforcement learning**   Several studies are based on the rationale that language models have already acquired comprehensive knowledge during the pre-training, and only high-quality supervised fine-tuning data is required for further tuning (Zhou et al., 2023). So these works (Liu et al., 2023b; Sun et al., 2023; Bai et al., 2022b; Bhardwaj & Poria, 2023; Krishna et al., 2022; Gulcehre et al., 2023) bypass reward modeling, and instead concentrate on the construction of datasets that align well with human preferences. Other works are directed towards exploring substitutes for the intricate PPO algorithm. These efforts employ diverse approaches to learn from the preference data, encompassing the creation of a supervised fine-tuning training dataset enriched with

human preference data (Liu et al., 2023a; Zhang et al., 2023; Dong et al., 2023), the integration of preferences for different outputs into the loss function (Yuan et al., 2023; Rafailov et al., 2023; Zhao et al., 2023b; Liu et al., 2023d;c), and the utilization of controllable text generation techniques (Lu et al., 2022). However, the human preference information used in these methods is at the sentence level, lacking more fine-grained supervision signals.

## 3 APPROACH

In this section, we present the proposed alignment approach **FIGA** by leveraging fine-grained quality signals. Our approach is developed based on a specially curated alignment dataset called *SPA* (Section 3.1), where each low-quality initial response is paired with a high-quality revised response. Based on such an alignment dataset, we further develop a new loss function that incorporates fine-grained quality signals derived by contrasting good and bad responses (Section 3.2). Our approach is easy to implement (similar to SFT) and can capture the underlying effect to generate high-quality responses instead of simply imitating them (similar to RLHF), which are discussed in Section 3.3. The overall framework of our FIGA pipeline is shown in Figure 1.

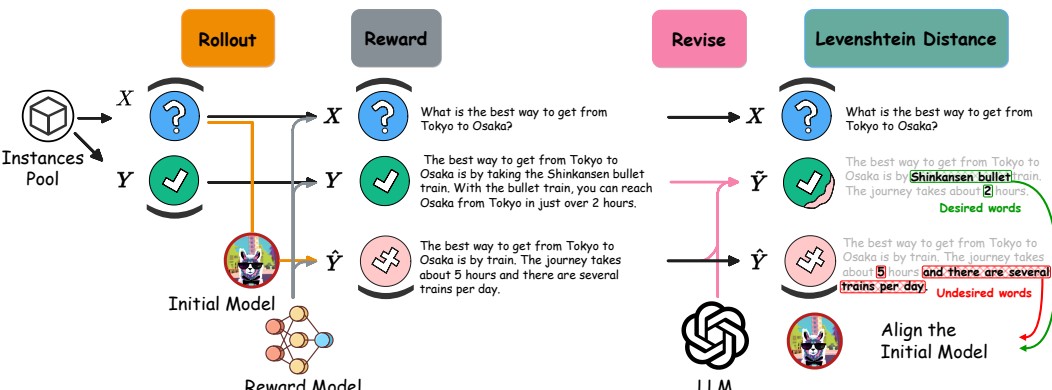

Figure 1: The overall illustration of our alignment approach FIGA.

### 3.1 CURATED ALIGNMENT DATASET

From the perspective of dataset, the novelty of our alignment approach can be given in two major aspects. Firstly, we don't directly aggregate all the available instruction data, but instead focus on high-quality instruction data that a LLM *performs less well* on. It enables LLMs to specially improves their weaknesses, reducing the cost of replicate learning. Secondly, we don't take what human annotators write or powerful LLMs (*e.g.,* ChatGPT or GPT-4) generate as training targets, but instead seek a more similar surrogate that is derived based on its own output by a LLM. It can largely reduce the distribution shift between the LLM to be aligned and the ground-truth demonstrations.

We carefully construct the **SubP**ar **A**lignment (SPA) dataset, a curated collection of query, model's initial response, and the corresponding improved response (with minor revision). Compared with prior work (Ouyang et al., 2022; Yuan et al., 2023; Liu et al., 2023a), we mainly consider the queries where LLMs' performance are not satisfactory and aim to correct these bad cases via specific training. Moreover, we refine the initial response of a LLM that is to be aligned as training target, which can effectively reduce the distribution shifts from the ground-truth demonstrations.

Formally, we denote the initial model as $\pi_\theta$, which can be a supervised-finetuned model (*e.g.,* Alpaca (Taori et al., 2023)) or a pre-trained base model (*e.g.,* LLaMA (Touvron et al., 2023a)). To construct our dataset, we assume that a reward model for assessing the alignment level is available. In practice, a number of reward models have been released publicly (*e.g.,* DeBERTa (OpenAssistant, 2023)), which can be used for our approach. Given a query $X$ and a response $Y$, we leverage a reward model RM to compute the reward score $R_Y = \text{RM}(X, Y)$, which reflects how well the response $Y$ aligns with given query $X$. Below, we detail the construction procedure.

**Rollout for initial response generation** We first broadly collect existing paired datasets encompassing a wide range of real-world tasks, and construct the instances pool $\mathcal{D} = \{X, Y\}_{i=1}^{n}$. To better align with human value, we select preference datasets (*e.g.,* HH-RLHF (Bai et al., 2022a)) that adhere to the 3H principle (*i.e.,* helpfulness, honesty, and harmlessness) in this work. Furthermore, we also include instruction dataset (*e.g.,* OpenOrca (Mukherjee et al., 2023)) to preserve the task solving abilities of LLMs. We aim to train a both capable and safe model like ChatGPT, rather than only focusing on alignment while sacrificing the task solving abilities. Based on these datasets, we employ the rollout model $\pi_\theta$ to generate initial responses $\hat{Y} = \pi_\theta(X)$ for the given queries.

**Identifying the queries to be enhanced** After obtaining the model's initial response $\hat{Y}$ and the human-preferred response $Y$, we next identify the queries where the model requires further improvement to better align with human intent through the reward score $\mathrm{RM}(\cdot)$. Following existing work (Ouyang et al., 2022), we employ the reward model as a surrogate of human preferences, and design a filtering process based on the calculated reward score $R_{\hat{Y}}$ and $R_Y$ for all the instances. We only keep the instances that meet all the three following restrictions: (1) $R_{\hat{Y}} < \eta_1$ (a subpar initial performance, *i.e.,* bad cases), (2) $R_Y > \eta_2$ (high-quality demonstrations), and (3) $R_Y - R_{\hat{Y}} > \eta_3$ (clear quality difference), where $\eta_1$, $\eta_2$, and $\eta_3$ are three threshold values for filtering, we will set them according to the reward score distribution. The details can be found in Section 4.1.2. With the above filtering mechanism, we ensure the quality and usefulness of our SPA dataset. We target at bad case correction of the rollout model, which is more directed and effective than existing methods that directly trains the model on the whole collected dataset.

**Revising initial responses for reducing the distribution shifts** To align a LLM, a basic principle is to ensure that the distribution of the model should not experience significant shifts during the alignment process (Bai et al., 2022a). Despite that the ground-truth demonstration ($Y_i$) is human preferred, it is likely to span a very different semantic distribution as the LLM to be aligned. Our solution is to revise the initial response ($\hat{Y}$) by referring to the ground-truth demonstration ($Y_i$). In this way, we can effectively reduce the distribution shifts as well as obtaining demonstrations similar to the original output. Specially, we generate a pseudo reference $\tilde{Y}$ based the target $Y_i$, making minor adjustments to the $\hat{Y}$ and enhance its quality, *i.e.,* modifying $\hat{Y}$ as minimally as possible based on $Y_i$. Such a generation process is conducted by prompting the powerful ChatGPT. To facilitate the generation process, we further manually inspect the low-quality responses that we have previously filtered and identify four major low-quality reasons: (1) lack of detail, (2) inaccuracy in response, (3) the need for structural adjustments, and (4) other factors (off-topic or harmful content). In detail, we leverage ChatGPT to determine, given $Y_i$, which of the four reasons $\hat{Y}$ is associated with. Afterwards, we design different prompts for the four reasons and instruct the LLM to make minor correction to the initial response $\hat{Y}$ based on $Y_i$. We denote the revised response as $\tilde{Y}$. The details of our process and prompts can be found in Appendix B.

Finally, we obtain the SPA dataset $\{X, \hat{Y}, \tilde{Y}\}$ for subsequent training. Our construction method has dual merits: it not only aligns the reference output with human preferences but also preserves the inherent linguistic style and overall semantic distribution of the model to be aligned. Note that we keep both the initial and revised responses in a contrastive form, because they are jointly used for deriving fine-grained quality signals in subsequent training.

## 3.2 FINE-GRAINED QUALITY-AWARE ALIGNMENT TUNING

As described above, our fine-tuning dataset for alignment contains both low-quality initial responses ($\hat{Y}$) and high-quality revised responses ($\tilde{Y}$). Instead of directly learning from these high-quality responses (similar to rejection sampling (Touvron et al., 2023b)), it is important for LLMs to understand why such revisions are useful to produce the high-quality responses. Furthermore, LLMs can improve the alignment capacity from the contrast between good and bad responses.

Motivated by previous work (Liu et al., 2022), we utilize Levenshtein distance to quantify the similarity between of $\hat{Y}$ and $\tilde{Y}$. Levenshtein distance is a dynamic programming algorithm to obtain the minimal edit distance between two sentences through three operations: addition, deletion, and substitution. Comparing the initial and revised response, the involving tokens can be generally divided into three types: newly added, deleted, or substituted. We consider assigning different weights to

these three types of tokens. We reward the tokens that are added or substituted in the revised response $\tilde{Y}$, penalize the tokens that are deleted or substituted in the original response $\hat{Y}$, and tend to overlook the rest tokens that remain the same after the revision process. Formally, we introduce two token-level weighting functions to characterize the above ideas:

$$\tilde{r}(\tilde{y}_t, t) = \begin{cases} \alpha, & \text{if } \tilde{y}_t \text{ is added or substituted} \\ \gamma, & \text{otherwise} \end{cases}, \hat{r}(\hat{y}_t, t) = \begin{cases} \beta, & \text{if } \hat{y}_t \text{ is deleted or substituted} \\ 0, & \text{otherwise} \end{cases}, \quad (1)$$

where $\alpha > 0$, $\beta > 0$, and $\gamma \geq 0$ are three coefficients to control the encouraged, discouraged, and ignored parts, which can be empirically set or learned from tuning data.

In this way, we can then encourage the model to "imitate" the desired actions that have a greater impact on enhancing quality, discourage the model from emulating the undesired actions that lead to a poor performance in quality. The final training loss can be formulated as:

$$\mathcal{L} = - \underbrace{\sum_{\tilde{y}_t \in \tilde{Y}} \tilde{r}(\tilde{y}_t, t) \log \pi_\theta(\tilde{y}_t | \tilde{y}_{<t}, X)}_{\text{increase the probability of desired words}} + \underbrace{\sum_{\hat{y}_t \in \hat{Y}} \hat{r}(\hat{y}_t, t) \log \pi_\theta(\hat{y}_t | \hat{y}_{<t}, X)}_{\text{decrease the probability of undesired words}}. \quad (2)$$

The overall FIGA pipeline is illustrated in Algorithm 1. The major advantages of FIGA over typical SFT (Ouyang et al., 2022) is that it can learn from fine-grained contrast between good and bad responses, which is essentially similar to that in reinforcement learning (discussed in Section 3.3). In addition, by explicitly modeling the revision effect, such an approach can naturally zoom into crucial words or phrase, making the model better zoom into fine-grained semantics.

---

**Algorithm 1:** FIGA - Leveraging Fine-grained Quality Signals for Alignment

1 **Input:** Instance pool $\mathcal{D} = \{X, Y\}_{i=1}^n$, initial model $\pi_\theta$, revision model (ChatGPT), reward function $R_{(\cdot)}$.
2 ### SPA Dataset Construction
3 **for** *each instance* $\{X, Y\}$ *in* $\mathcal{D}$ **do**
4      1. Rollout for initial generation. Generate $\hat{Y} \sim \pi_\theta(X)$ and compute $R_Y, R_{\hat{Y}}$;
5      2. Reward filtering. **if** $R_{\hat{Y}} > \eta_1$ *or* $R_Y < \eta_2$ *or* $R_Y - R_{\hat{Y}} < \eta_3$ **then**
6          Discard the current instance;
7      3. Response Revision. Analyze the reason for the poor performance of $\hat{Y}$, and generate the corresponding revision $\tilde{Y} \sim \text{LLM}(\hat{Y}, Y)$ based on the identified reason.
8 Construct the SPA dataset $\mathcal{S} = \{X_i, \hat{Y}_i, \tilde{Y}_i\}_{i=1}^m$.
9 ### Alignment Learning
10 **for** *epoch* $e = 1, ..., E$ **do**
11      **for** *each instance* $\{X, \hat{Y}, \tilde{Y}\}$ *in SPA* $\mathcal{S}$ **do**
12          Locate the crucial parts with Levenshtein distance using Equation 1 and assign weights according to $\tilde{r}(\tilde{y}_t, t)$ and $\hat{r}(\hat{y}_t, t)$;
13          Update $\pi_\theta$ using the fine-grained quality-aware learning objective in Equation 2.

---

### 3.3 DISCUSSION

In this part, we discuss how the proposed FIGA approach relates to existing fine-tuning approaches, namely SFT and RLHF.

**Relationship with SFT**    SFT can be viewed as a special case of our FIGA method without revision, when training is performed with the higher-quality instance $Y$, and each token of $Y$ is considered equally important. Compared to SFT, FIGA has the following two advantages: (1) we only consider the inferior part of the bad case that the initial model does not perform well; (2) we explicitly enforce the model to understand what are good and bad behaviors in the loss function. It inherits the merits of SFT, and further leverages fine-fined quality signals for improving the alignment.

**Relationship with RL**    Our method can be considered as a simplified but efficient version of RL. Using typical PPO method (Schulman et al., 2017) as an example, its objective is to optimize the actor model (*i.e.,* the initial model $\pi_\theta$) to maximize the expected reward score, formally given as:

$$\mathcal{L}^{PPO} = -\sum_t \left( \frac{\pi_\theta(\hat{y}_t|\hat{y}_{<t}, X)}{\pi_{\theta_{\text{old}}}(\hat{y}_t|\hat{y}_{<t}, X)} \cdot A_{\hat{y}_t} \right), \tag{3}$$

where $A_{\hat{y}_t}$ is the advantage function of the $\hat{y}_t$ token returned by the critic model given the reward score $R_{\hat{Y}}$. $\pi_{\theta_{\text{old}}}$ is the model before the previous parameter update. Here, we ignore the clipping function and KL penalty for convenience. Considering the FIGA training objective in Equation 2, our weight functions $\tilde{r}(\cdot)$ and $\hat{r}(\cdot)$ in FIGA can be viewed as a simplified advantage function $A_{(\cdot)}$ in Equation 3 to evaluate the importance of each token. Therefore, FIGA has a similar objective with RL but with a simplified token-wise reward function. We do not use an extra learned critic model and remove the use of previous rollout model, which makes FIGA more efficient. In the later experiment section, we will verify the effectiveness of our method.

## 4 EXPERIMENT

### 4.1 EXPERIMENTAL SETUP

#### 4.1.1 BASELINE METHODS

In order to better evaluate the FIGA method, we choose several baselines for comparison: (1) SFT (Ouyang et al., 2022): it continues to fine-tune the initial model using pairs of data with sequence-to-sequence loss. (2) PPO (Ouyang et al., 2022): it optimizes the initial model to achieve a higher reward score provided by the reward model. (3) CoH (Liu et al., 2023a): it annotates the dataset by prefixing "*A helpful answer:* " and "*An unhelpful answer:* " to the responses of corresponding quality, employs SFT on it, and computes loss only for the specially masked tokens. (4) RRHF (Yuan et al., 2023): it applies SFT on the optimal responses and further optimizes the ranking loss among responses from multiple sources to encourage the model to achieve a greater log probability for the response that ranks better. (5) DPO (Rafailov et al., 2023): it eliminates the need for explicit reward modeling and instead directly optimizes the policy model using comparison data.

#### 4.1.2 IMPLEMENTATION DETAILS

**Training Datasets**    For our SPA dataset mentioned in Section 3.1, we broadly select the following datasets as our initial instance pool: HH-RLHF (Bai et al., 2022a), ShareGPT (ShareGPT, 2023), Instruct GPT-J Pairwise (Dahoas, 2023), SHP (Ethayarajh et al., 2022), and OpenOrca (Lian et al., 2023). We employ Alpaca-7b Taori et al. (2023) as the rollout model to generate responses $\hat{Y}$ and use `gpt-3.5-turbo` to revise and obtain $\tilde{Y}$. The prompt used here can be found in Appendix B. As for the filtering process, we utilize `OpenAssistant/reward-model-deberta-v3-large-v2` (OpenAssistant, 2023) as the reward model. According to reward score distribution 2, we empirically set the threshold values $\eta_1 = 1, \eta_2 = 3, \eta_3 = 3.5$, respectively. The statistics for reward scores and edit operations of the SPA dataset are presented in Table 1, and the graphical illustration of reward scores is provided in Figure 2. We find that initial responses $\hat{Y}$ exhibit a large distributional disparity compared with the reference responses $Y$, which may complicate the learning process for the model. In contrast, our modified responses not only align more closely with the original distribution but also enhance the quality, which simplifies the learning task for the rollout model. The completed SPA dataset consists of 17,333 instances, and more details and analysis can be found in Appendix D.

**Model Details**    (1) For SFT, we set the learning rate to 1e-5 and the batch size to 128. We conduct 5 epochs of training and choose the one with the highest reward score on the test set as the ultimate SFT model. (2) For PPO, we apply the OpenLLaMA2 (OpenLLMAI, 2023) library and adhere to its hyper-parameter configurations. We use Alpaca-7b as the initial critic model and use the same reward model utilized in SPA construction. Given the modest gains observed in previous experiments when employing PPO-ptx on models with around 6B parameters (Ouyang et al., 2022), we refrain from introducing a pre-training mix as an additional training objective. (3) For CoH, we annotate the SPA dataset with their method. Considering the smaller size of our dataset compared to theirs, we set FCM (random masked token ratio to prevent overfitting) to 0. Additionally, to ensure a fair comparison with PPO, we disable the pre-training dataset regularization. (4) For RRHF and DPO, we follow the recommended hyper-parameters from the original papers. (5) For FIGA, we

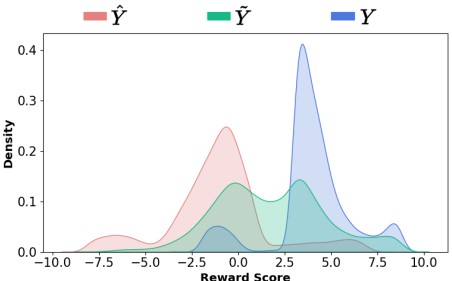

Figure 2: Reward score distributions.

Table 1: The average reward score of response data and the average number #ops of editing operations to them from the $\hat{Y}$.

| Data | $\hat{Y}$ | $Y$ | $\tilde{Y}$ |
|---|---|---|---|
| $R_{(\cdot)}$ | -1.07 | 3.94 | 1.78 |
| #ops | – | 75.69 | 39.38 |

set the parameters $\alpha = 1, \beta = 0.5, \gamma = 0$ respectively. Besides, considering the instability when training on negative samples in practice (Bhardwaj & Poria, 2023; Liu et al., 2023a), we further select the bad tokens returned by Levenshtein distance in equation 1 by retaining only those with a negative log-likelihood less than 0.6.

### 4.1.3 EVALUATION TASKS

We evaluate the performances of different methods on comprehensive benchmarks. We segment a test set from the selected datasets and utilize the reward score to evaluate how effectively the model has learned to align with human preferences. The resulting test set comprises a total of 3,608 data entries. Additionally, we employ a broad array of out-of-distribution benchmarks to conduct a more comprehensive evaluation of the model's capabilities. This includes assessing knowledge utilization (MMLU (Hendrycks et al., 2020)), human alignment (WinoGender (Rudinger et al., 2018), CrowS-Pairs (Nangia et al., 2020), and TruthfulQA (Lin et al., 2021)), and open-ended generation (Vicuna (Chiang et al., 2023) and WizardLM (Xu et al., 2023)). The details of evaluation tasks can be found in Appendix C.

### 4.2 EXPERIMENTAL RESULTS

Table 2: Performance comparison of FIGA and other widely used alignment methods. Bold and underlined fonts indicate the best and the second-best score. ↓ denotes lower is better.

| Methods | Reward | MMLU | TruthfulQA | CrowS-Pairs↓ | WinoGender | Vicuna | WizardLM | Average[1] |
|---|---|---|---|---|---|---|---|---|
| **Alpaca-7b** | 3.96 | 39.2 | 33.7 | 61.1 | 55.6 | 7.9 | 7.0 | 31.7 |
| **SFT** | 4.56 | 39.3 | 22.0 | 61.5 | 55.3 | 8.4 | **8.3** | 31.1 |
| **PPO (SPA)** | 4.06 | 39.6 | 30.1 | 61.3 | 56.2 | 7.6 | 7.4 | 31.5 |
| **PPO (85K)[2]** | 4.54 | 39.2 | 36.7 | 60.6 | 56.2 | 7.9 | 7.2 | 33.1 |
| **CoH** | 4.24 | 39.6 | 28.2 | **59.6** | 52.1 | 8.3 | 8.1 | 32.7 |
| **RRHF** | 4.23 | 37.8 | 32.9 | 59.9 | **60.0** | 7.9 | 7.9 | 31.3 |
| **DPO** | 4.23 | 40.1 | 34.8 | 61.2 | 57.0 | 8.0 | 7.7 | 32.7 |
| **FIGA** | **4.62** | **40.8** | **42.0** | 61.2 | 59.6 | **8.6** | **8.3** | **34.9** |

As in Table 2, FIGA surpasses all baselines, showing superior performance across benchmarks, even outperforming PPO using four times training data. This implies FIGA aligns more closely with human preferences and exhibits strong overall task-solving capabilities.

Moreover, to assess the comparative advantages of each response, we conduct a comparison between the responses generated by FIGA and other baseline methods on the Vicuna and WizardLM benchmarks. The results are shown in Figure 3. And we also conduct human evaluation in Appendix F for more fine-grained analysis.

---

[1]To reflect the model's overall performance, we compute the average score. Specifically, we multiply the reward score by 10, and the score for CrowS-Pairs is calculated as 100 minus the original score.

[2]Given that PPO does not utilize labels in the dataset and requires a large amount of data to learn through trial and error, we integrate additional open-source data with the SPA dataset to fully leverage the strengths of PPO. We obtain a total of 84,908 entries, and the PPO trained with this dataset is referred to as PPO (85K).

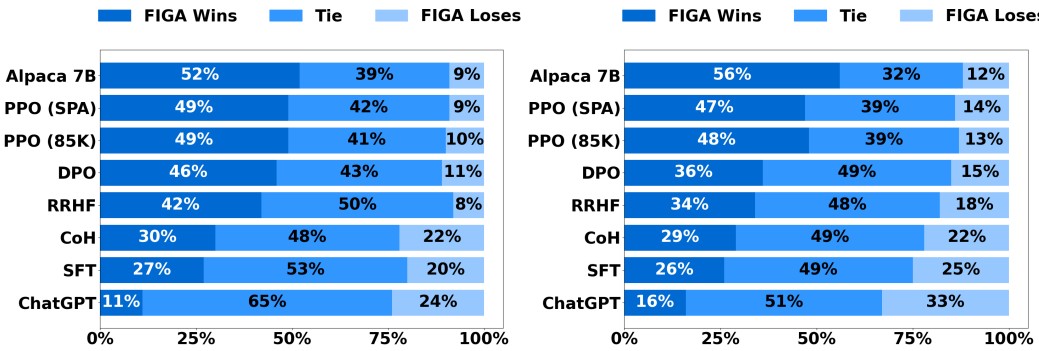

Figure 3: Win rate of FIGA vs other baselines on Vicuna (left) and WizardLM (right).

## 4.3 FURTHER ANALYSIS

### 4.3.1 PERFORMANCE COMPARISON W.R.T. SUBPAR ALIGNMENT DATASET

As mentioned in Section 3.1, the steps involved in constructing the SPA dataset include: (1) collecting existing datasets, including preference datasets and typical instruction datasets; (2) filtering the data based on reward scores; and (3) revising the initial responses using LLM. To examine the effectiveness of each of them, we develop the following dataset variants:

- **Preference**: we only use preference data to construct the initial instance pool $\mathcal{D}$ with 3,971 samples.
- **Instruction**: we construct the initial instance pool $\mathcal{D}$ with typical instruction data that the reward model had not encountered during its training, totaling 3,971 instances.
- **W/o reward filtering**: this variant excludes the step of data filtering according to reward scores.
- **W/o revision**: we do not utilize LLM to revise and instead directly employ the reference responses.

Table 3: Performance comparison of different instances pools.

| Methods | Reward | MMLU | TruthfulQA | CrowS-Pairs↓ | WinoGender | Vicuna | WizardLM | Average |
|---|---|---|---|---|---|---|---|---|
| **Preference** | **4.42** | 37.4 | 22.6 | 61.5 | 57.1 | 7.4 | 6.6 | 30.5 |
| **Instruction** | 4.35 | **40.7** | **31.1** | **59.7** | **57.5** | **8.5** | **8.2** | **32.8** |

Table 4: Performance comparison of different data annotations.

| Methods | Reward | MMLU | TruthfulQA | CrowS-Pairs↓ | WinoGender | Vicuna | WizardLM | Average |
|---|---|---|---|---|---|---|---|---|
| **FIGA** | **4.62** | **40.8** | **42.0** | 61.2 | **59.6** | **8.6** | **8.3** | **34.9** |
| **W/o reward filtering** | 4.41 | 38.0 | 28.8 | **61.1** | 58.5 | 8.3 | 8.0 | 32.1 |
| **W/o revision** | 4.39 | 37.5 | 26.7 | 62.1 | 55.6 | 8.2 | 7.7 | 31.1 |

From the results in Table 3 and Table 4, we can see that: (1) FIGA demonstrates strong performance on typical instruction data that is new to the reward model, proving that its applicability is not restricted to preference data. (2) Filtering based on reward scores is crucial, resulting in a +0.21 reward score increase and a +2.8 benchmark increase. This underscores the significance of training on queries where the model's original performance is subpar. (3) Addressing the distribution shift through revisions is important, as training with revisions yields +3.8 points on average.

### 4.3.2 PERFORMANCE COMPARISON W.R.T. WEIGHTING FUNCTIONS

As mentioned in Section 3.2, $\tilde{r}(\cdot)$ and $\hat{r}(\cdot)$ in Equation 1 first make comparison between $\hat{Y}$ and $\tilde{Y}$, and then assign distinct weights to various tokens. Here, we explore other weighting functions as how they acquire the tokens to be encouraged or discouraged, and study the influence of different hyper-parameters ($\alpha$, $\beta$, and $\gamma$). More details on hyper-parameters can be referred to in Appendix E.

- **Variants of $\tilde{r}(\cdot)$**: we set $\beta$ to 0 and propose three different variants to explore alternative methods for identifying the tokens that should be encouraged.

  - **Bag of words**: it sets $\tilde{r}(\tilde{y}_t, t) = 1$ only when $\tilde{y}_t \notin \hat{Y}$; while the rest are set to 0.
  - **ChatGPT (weighted)**: motivated by the work (Lee et al., 2023), it employs ChatGPT to assess the impact of tokens on sentence quality. The specific prompt can be found in B. The returned scores are adjusted to fall within the range of 0.7 to 1.3 and are set as $\tilde{r}(\tilde{y}_t, t)$. For words that ChatGPT doesn't address, $\tilde{r}(\tilde{y}_t, t) = 0.3$.
  - **ChatGPT (binary)**: it sets $\tilde{r}(\tilde{y}_t, t)$ to 1 only when $\tilde{y}_t$ is returned by ChatGPT with a non-zero score, while the rest are set to 0.

- **Variants of $\hat{r}(\cdot)$**: as for the tokens to be discouraged returned by $\hat{r}(\cdot)$, we further filter bad tokens returned by Levenshtein distance and retain only those with a negative log-likelihood below 0.6. To assess its effectiveness, we design the variants including:

  - **Inverted threshold**: it retains only the bad tokens returned by Levenshtein distance with a negative log-likelihood $\geq 0.6$.
  - **W/o further selection**: it penalizes all the bad tokens returned by Levenshtein distance.

- **Variants of hyper-parameters**: to explore the influence of $\alpha, \beta, \gamma$ in Equation 1, we design:

  - **$\beta = 0$**: it sets $\beta$ to 0 with $\alpha = 1$ and $\gamma = 0$.
  - **$\gamma \neq 0$**: it sets $\gamma$ to 0.3 with $\alpha = 1$ and $\beta = 0.5$.
  - **$R_{(\cdot)}$**: it assigns $R_{\tilde{Y}}, R_{\hat{Y}}, 0$ to $\alpha, \beta, \gamma$ respectively, where $R_{\tilde{Y}}$ and $R_{\hat{Y}}$ are standardized through the min-max method.

Table 5: Performance comparison of different weighting functions.

| Explorations | Methods | Reward | MMLU | TruthfulQA | CrowS-Pairs↓ | WinoGender | Vicuna | WizardLM | Average |
|---|---|---|---|---|---|---|---|---|---|
| **Ours** | FIGA | **4.62** | **40.8** | **42.0** | 61.2 | **59.6** | **8.6** | **8.3** | **34.9** |
| **Encouraged** | Bag of words | 4.52 | 40.4 | 29.3 | 60.0 | 57.6 | 8.1 | 8.2 | 32.7 |
| | ChatGPT (weighted) | 4.37 | 39.8 | 21.7 | 60.0 | 57.9 | 8.4 | 8.1 | 31.4 |
| | ChatGPT (binary) | 4.32 | 39.0 | 24.4 | 59.9 | 59.0 | 7.8 | 7.6 | 31.6 |
| **Discouraged** | Inverted threshold | 3.80 | 30.2 | 27.2 | **56.2** | 50.4 | 8.1 | 7.4 | 29.3 |
| | W/o further selection | 3.01 | 28.1 | 24.0 | 58.5 | 57.4 | 8.0 | 7.7 | 28.1 |
| **Hyper-parameter** | $\beta = 0$ | 4.61 | 41.0 | 37.0 | **59.6** | 58.1 | 8.5 | 8.3 | 34.2 |
| | $\gamma \neq 0$ | 4.54 | **41.2** | 32.2 | 60.1 | 56.0 | 8.4 | 8.2 | 33.0 |
| | $R_{(\cdot)}$ | 4.54 | 39.7 | 37.8 | 62.9 | 57.1 | 8.2 | 8.2 | 33.4 |

The results in Table 5 indicate that: (1) Levenshtein distance excels in extracting critical tokens, with over +1.5 and +2.6 average scores compared with the statistical method and ChatGPT annotation method. (2) It is necessary to further filter the bad tokens returned by Levenshtein distance, as this leads to an average improvement of +6.8. (3) Remaining only the poor-quality tokens with a negative log-likelihood $\leq 0.6$ is a sensible choice, which aims to penalize tokens that the model is relatively confident in generating, even though their actual quality is subpar. (4) Punishing the undesirable actions is beneficial, as it results in an average increase of +0.7. (5) Focusing only on good and bad tokens is sufficient, since setting $\gamma$ to a non-zero value leads to a decrease of 1.9. (6) The inferior performance of reward score weights can be attributed to intrinsic inaccuracies of the reward scores, especially in out-of-distribution scenarios (Bai et al., 2022b).

## 5  CONCLUSION

In this paper, we have presented FIGA, a new approach that aligns language models with human preferences, by leveraging fine-grained quality signals to enhance the alignment quality during fine-tuning. In our approach, we firstly curate a high-quality alignment dataset that pairs initial responses with revised responses on queries that a LLM cannot perform well. Furthermore, we have designed a new learning objective that that can leverage the fine-grained quality signals by contrasting initial with revised responses. Our approach inherits the merits of SFT (*e.g.,* efficient and easy-to-implement), and meanwhile can better understand and learn what are correct behaviors for alignment. FIGA shows superior performance on extensive tasks, with +3.2 points and +1.8 points against the initial supervised-finetuned model and the strong PPO method.

## ACKNOWLEDGMENTS

This work was partially supported by National Natural Science Foundation of China under Grant No. 62222215, Beijing Natural Science Foundation under Grant No. L233008 and 4222027. Xin Zhao is the corresponding author.

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

## A   APPENDIX: DATA SOURCES

(1) **HH-RLHF (Helpful and Harmless)**: It comprises two main categories of data: human preference data about helpfulness and harmlessness, and human-annotated red teaming dialogues. The first one is pivotal to train the reward model, and the second one gives insights into model red-teaming techniques[1].

(2) **ShareGPT**: The dataset contains conversations through the API using process. Within each conversation, both user prompts and ChatGPT responses from OpenAI are presented[2].

(3) **Synthetic Instruct GPT-J Pairwise**: Crafted for instruction-oriented tasks, it explores model-generated outputs when exposed to synthetic prompts[3].

(4) **Stanford SHP**: It offers 385K human preferences across multiple disciplines. These preferences are designed to discern the relative helpfulness of responses. Contrary to the HH-RLHF dataset, all content in SHP is penned by humans, serving as a valuable complement to other datasets[4].

(5) **OpenOrca**: This dataset is an extension of the FLAN Collection, including GPT-4 and GPT-3.5 model completions. Its primary application lies in training and evaluation in the realm of NLP. For our investigation, we've exclusively focused on the English instruction subset[5].

## B   APPENDIX: PROMPTS USED FOR DATA AUGMENTATION

**Details for revision**   Given a question, along with the poorer original model response and a preferred ground truth response, we instruct ChatGPT to make minimal modifications to the original response, while ensuring that the output still remains closely aligned with the preferred response.

This process can be divided into two steps: first analyzing the reasons for the lower quality of the original response based on the comparison, and then, making revisions using the appropriate prompts based on these factors.

Prompt to used analyze the reason: *Question: ... Response 1: ... Response 2: ... Among them, the quality of Response 1 is inferior to that of Response 2. Please compare them and choose one of the following four possible reasons for the area where Response 1 performed the worst: A. Needs more accurate content, B. Needs more comprehensive content or more details, C. Requires adjustments in structure, D. Other reasons (such as containing harmful information or going off-topic). Do not include analysis, but just return the choice.*

[1] https://huggingface.co/datasets/Anthropic/hh-rlhf
[2] https://huggingface.co/datasets/anon8231489123/ShareGPT_Vicuna_unfiltered
[3] https://huggingface.co/datasets/Dahoas/synthetic-instruct-gptj-pairwise
[4] https://huggingface.co/datasets/stanfordnlp/SHP
[5] https://huggingface.co/datasets/Open-Orca/OpenOrca

The prompts used to revise according to different reasons:

Prompt for reason A: *Question: ... Response 1: ... Response 2: ... Please replace the content corresponding to Response 1 with the accurate and high-quality essence from Response 2, and remain the original structure of Response 1. Ensure that the edit distance between the optimized Response 1 and the Response 1 is as low as possible.*

Prompt for reason B: *Question: ... Response 1: ... Response 2: ... Please incorporate the comprehensive topic or the details from Response 2 into Response 1, or if necessary, replace any synonymous content from Response 1 with that from Response 2. You must remain the original structure of Response 1, ensure the edit distance between the optimized Response 1 with the Response 1 is as low as possible, and not add new contents other than those contained in Response 1 and Response 2.*

Prompt for reason C: *Question: ... Response 1: ... Response 2: ... The structure of Response 2 is well-organized, featuring elements including but not limited to: 1. point-by-point addressing, 2. providing an overview of the question before answering. Use the structure of Response 2 to rephrase Response 1. Ensure that the optimized Response 1 should maintain a relatively low edit distance from the original Response 1.*

**Annotate the importance of each word** Given a question, along with the lower-quality original response from the initial model and a higher-quality ground truth response, we require ChatGPT to score each word based on comparison, in terms of how much it improve the quality. Below is an example.

*Below is an instruction that describes a task, followed by an original response and a better response in terms of how well it aligns with human preferences, being helpful, harmless, and honest. Your task is to return a list containing tuples with words and corresponding scores, which are meant to measure the extent to which the words improve the quality of the original answer to the better answer. The scores are all integers, with 0 being the lowest score and 5 being the highest score. Instruction: ... Original Response: ... Better Response: ...*

## C  APPENDIX: EVALUATION DETAILS

We utilize a suite of benchmarks to evaluate the performance as mentioned in Section 4.1.3. Here, we provide details on how we use these benchmarks for evaluation.

**MMLU:** The multitask multilingual language understanding (MMLU) benchmark consists of 57 subtasks. Our approach involves calculating the negative log-likelihood of the correct option across all tasks, ensuring it is the minimal among all options. Finally, we calculate the accuracy of correctly answered questions. The implementation details can be found at the *LM-evaluation-harness* repository [6].

**TruthfulQA:** For the TruthfulQA benchmark, we select the multiple-choice task (MC1), which contains only one correct option. Similar to MMLU, we evaluate by determining if the negative log-likelihood of the correct option is the lowest among all options, and calculate the accuracy of correctly answered questions.

**CrowS-Pairs:** This benchmark involves comparing two sentences: one exhibits a common form of bias, while the other one does not, differing only in a few words. We calculate the perplexity for each sentence and determine the bias rate by evaluating which sentence has a lower perplexity. A bias rate closer to 50% indicates less bias in the model.

**Winogender:** It assesses the model's accuracy in handling Winograd schema challenge tasks under three scenarios: male, female, and unspecified gender. It is based on whether the negative log-likelihood of the correct sentence is lower. Our primary metric is the average score across these gender scenarios.

---

[6]https://github.com/EleutherAI/lm-evaluation-harness

**Vicuna and WizardLM:** These two benchmarks instruct the model to generate responses to the given prompts. The responses are then rated by ChatGPT (Zheng et al., 2023) on a scale of 1 to 10, with the overall performance measured by the average score.

## D APPENDIX: SPA DATASET DETAILS

We have conducted a budget estimation based on the usage of the GPT-3.5-turbo API for our SPA dataset. The average input token count per data entry in the SPA dataset is 952, and the average output token count is 143. Considering the pricing of API usage (*i.e.,* \$0.001 and \$0.002 per 1K tokens for input and output respectively), the total estimated cost for the entire SPA dataset amounts to \$21.45.

Additionally, we conduct experiments with various sizes of the SPA dataset to investigate how much data is necessary to achieve a reasonable performance gain. The results in Table 6 indicate that constructing a SPA dataset with 8000 samples and training based on the FIGA method can lead to a substantial performance boost.

Table 6: Performance comparison of different data sizes.

| #Data | Reward | MMLU | TruthfulQA | CrowS-Pairs↓ | WinoGender | Vicuna | WizardLM | Average |
|---|---|---|---|---|---|---|---|---|
| **Alpaca-7b** | 3.96 | 39.2 | 33.7 | 61.1 | 55.6 | 7.9 | 7.0 | 31.7 |
| **4k** | 4.28 | 39.9 | 26.3 | 59.0 | 55.7 | 8.6 | 8.4 | 31.8 |
| **8k** | 4.58 | 41.0 | 33.2 | 59.8 | 57.2 | 8.4 | 8.2 | 33.4 |
| **17k** | 4.62 | 40.8 | 42.0 | 61.2 | 59.6 | 8.6 | 8.3 | 34.9 |

## E APPENDIX: HYPTER-PARAMETER SETTINGS

Below are the results of our experiments conducted with various hyper- parameter configurations.

Table 7: Performance comparison of different hyper-parameters.

| Explorations | $\alpha/\beta/\gamma$ | Reward | MMLU | TruthfulQA | CrowS-Pairs↓ | WinoGender | Vicuna | WizardLM | Average |
|---|---|---|---|---|---|---|---|---|---|
| **Ours** | 1/0.5/0 | **4.62** | **40.8** | **42.0** | 61.2 | **59.6** | **8.6** | **8.3** | **34.9** |
| **w.r.t. $\alpha$** | $R_{(.)}/0.5/0$ | 4.54 | 39.7 | 37.8 | 62.9 | 57.1 | 8.2 | 8.2 | 33.4 |
| | $R_{(.)}/0/0$ | 4.46 | 39.9 | 27.3 | 60.0 | 60.1 | 8.1 | 7.9 | 32.6 |
| **w.r.t. $\beta$** | 1/0.5/0 | 4.65 | 41.1 | 41.4 | 60.1 | 57.1 | 8.6 | 8.4 | 34.7 |
| | 1/0.25/0 | 4.59 | 38.8 | 40.3 | 61.5 | 60.6 | 8.3 | 7.9 | 34.3 |
| | 1/0.2/0 | 4.64 | 40.1 | 38.3 | 61.7 | 59.9 | 8.4 | 8.3 | 34.2 |
| **w.r.t. $\gamma$** | 1/0.5/0.3 | 4.54 | 41.2 | 32.2 | 60.1 | 56.0 | 8.4 | 8.2 | 33.0 |
| | 1/0/0.3 | 4.50 | 40.4 | 30.8 | 59.8 | 56.4 | 8.4 | 8.1 | 32.8 |

From the Table 7, it can be observed that: (i) $\gamma$ should be set to 0, as a non-zero $\gamma$ leads to a decrease in results. (ii) $\alpha$ should be set to 1, as setting $\alpha$ to the reward score results in a decrease in performance. (iii) The final results are insensitive to $\beta$, indicating that as long as $\beta$ is set to a reasonable value, it will not notably impact the overall performance.

## F APPENDIX: HUMAN EVALUATION

In addition to all the automatic evaluation metrics, we further carry out human evaluation to test how much the model align to human preference after FIGA training. To be specific, we explore the following aspects:

Firstly, we randomly select 20 instructions from the test set and choose the outputs before FIGA training and after FIGA training. We ask 3 participants to assess whether these responses exhibit the four aforementioned types of errors. The results in Table 8 show that after FIGA training, errors of all types have significantly decreased.

Then, we evaluate the quality of outputs through human evaluation. We randomly select 30 instructions from Vicuna benchmark, along with corresponding responses from FIGA and other baseline

Table 8: Human evaluation of error types.

| Error Type | Inaccuracy | Lack of Details | Structure | Others (off-topic or harmful) |
|---|---|---|---|---|
| before FIGA training | 20% | 35% | 5% | 0% |
| after FIGA training | 5% | 10% | 0% | 0% |

methods (including ChatGPT). We invite three evaluators to make pairwise comparisons between FIGA responses and those of each baseline method, aiming to identify the one with higher quality. Finally, we aggregate the assessments from the three participants and obtain the final evaluation results through the mode method. The results in Table 9 show that FIGA outperforms all baseline methods, indicating its ability to better align with human preferences.

Table 9: Human evaluation of overall quality.

| FIGA against | Alpaca-7b | SFT | PPO | PPO (85k) | RRHF | CoH | DPO | ChatGPT |
|---|---|---|---|---|---|---|---|---|
| FIGA win rate | 73% | 57% | 73% | 83% | 73% | 57% | 73% | 40% |

