# OpenReview forum: "Beyond Imitation: Leveraging Fine-grained Quality Signals for Alignment"
_ICLR.cc/2024/Conference — ICLR 2024 poster_

### Official Review · Reviewer_hMwD · 2023-10-13

**Soundness:** 2 fair
**Presentation:** 3 good
**Contribution:** 2 fair
**Rating:** 6
**Confidence:** 3

**Summary:**

The paper proposes a novel alignment strategy as a replacement of the RL step in RLHF. This is motivated by the observed unstabilities in PPO, and the complexities involved in online optimization. The idea is to mitigate the limitations from behavior cloning by proposing expert annotations that are actually generated by GPT using the generations of the agent as a form of conditioning. Moreover, the training samples are selected thanks to an external reward model. They also include additional information during the supervised fne-tuning by over-weighting the key modifications/deletions. The proposed FIGA improves performances compared to baselines. They ablate the performances.

**Strengths:**

- Human aligned AI is a critical topic, thus considering better alignment strategies is interesting, in particular if we could make the RLHF more stable.
- Fine-grained reward is also a promising idea.
- The idea of modifying the agent predictions with an external model (rather than taking directly the human-preferred generation) makes sense, and seems interesting to limit the drawbacks from behaviour cloning.
- The procedure is siple, ad ths could be used in real-world applications.
- The writing is very clear, and the writing is good.

**Weaknesses:**

- The proposed strategy actually requires both (1) a reward model and (2) a preference dataset. In contrast, most other strategies require only one of those. Actually, the empirical gains can come from the fact that you optimize those two guide the learning.
- The approach requires to set many new different hyperparameters (6 I believe), reducing the generality of the strategy.
- FIGA also requires an external model to be available, such as gpt-3.5-turbo. This is a huge limitation. Moreover, this should be ablated against other strategies that would also use gpt-3.5-turbo, such as distillation strategies.
- More generally, the experiments lack many important baselines, in particular DPO [Rafailov2023]. I list other below, perhaps less critical and more recent, that adding them would be a big positive point.

[Rafailov2023] Direct Preference Optimization: Your Language Model is Secretly a Reward Model \
[Gulcehre2023] Reinforced Self-Training (ReST) for Language Modeling \
[Zhao2023] SLiC-HF: Sequence Likelihood Calibration with Human Feedback \
[Liu2023b] statistical rejection sampling improves preference optimization \
[Peng2023] stabilizing rlhf through advantage model and selective rehearsal \

- Minors
    * There is a typo in the legend of Figure 1, the main Figure of the paper.
    * As illustrated in Algorithm 1, the proposed strategy remains essentially offline, as the SPA dataset is built once. One solution would be to build the dataset at each epoch, as done in [Gulcehre2023].
    * The experiments are only made for Alpaca-7b, which may not be illustrative of larger models.

**Questions:**

- Rather than using Levenshtein distance, I am interested in knowing whether you could directly ask gpt-3.5-turbo to explicitly state which part of the sentence is correct/false.
- Moreover, to further improve results, that would be interesting to give to gpt-3.5-turbo both the human-preferred generation along with the agent generation. Somehow, gpt-3.5-turbo could keep high rewards while reducing the #ops.

---

> ### Author Response · Authors · 2023-11-17
> **1. Requirement of both RM and preference data**
>
> Thanks for your detailed and meaningful comments! We have listed our response to your concerns as follows. If you also have any other questions, please feel free to let us know. We will continue to try our best to answer for you.
>
> **1. The proposed strategy actually requires both (1) a reward model and (2) a preference dataset. In contrast, most other strategies require only one of those. Actually, the empirical gains can come from the fact that you optimize those two guide the learning.**
>
> (1) First, **preference data is not essential for the FIGA method**, as demonstrated by the experiments in Table 4. If initially using only conventional instruction data without any preference data, a good result can still be obtained based on the FIGA method.
>
> (2) Besides, it is important to clarify the role that, the **reward model serves only for identifying weaknesses in the initial model and selecting high-quality ground truths for constructing the SPA dataset**. This process is essentially about data quality filtering, i.e. the creation of the SPA dataset. During the training phase, we do not employ the reward model, so the training phase of FIGA is both memory-friendly and efficient.
>
> (3) Furthermore, we do not consider the advantage of FIGA stems largely from using RM and preference data. **Considering that in our baseline comparisons (Table 2 in our paper), other methods also incorporate the SPA dataset**, which also uses a reward model for dataset construction and inherent preference data. Despite this, FIGA maintains a significant lead, underscoring the effectiveness of FIGA's training objective beyond the mere use of these two elements.

---

> ### Author Response · Authors · 2023-11-17
> **2. Many hyper-parameters**
>
> **2. The approach requires to set many new different hyperparameters (6 I believe), reducing the generality of the strategy.**
>
> (1) **In reality, we don't have a multitude of parameters to tune**. The current hyperparameters are included for the sake of method completeness and clear exposition. Setting $\alpha=1$ and $\gamma=0$ is a natural choice since we need to emphasize positive words and ignore those that would naturally occur. For the other three parameters $\eta_1, \eta_2, \eta_3$, they are set to filter out the queries for which the model needs further improvement. In practice, there is no need for careful selection, and we chose them based on the reward scores distribution in Figure 2.
>
> (2) To further illustrate, we conducted experiments (which are also shown in Table 5 in our paper) demonstrating that these **hyperparameters are not sensitive**.
>
> | $\alpha / \beta / \gamma$ | Reward | MMLU | TruthfulQA | CrowS-Pairs | Vicuna | WizardLM | WinoGender | Average  |
> | :-----------------------: | :----------------------: | :--: | :-------------: | :----------: | :----: | :------: | :--------: | :--: |
> |       (FIGA) 1/0/0        |           4.61           |  41.0  |       37.0        |     59.6     |  8.5   |  8.3   |    58.1    | 34.2 |
> |      w.r.t. $\gamma$      |                          |      |                 |              |        |          |            |      |
> |        1/0.5/0.3         |           4.54           | 41.2 |      32.2       |     60.1     |  8.4   |   8.2    |     56.0     |  33.0  |
> |          1/0/0.3          |           4.50            | 40.4 |      30.8       |     59.8     |  8.4   |   8.1    |    56.4    | 32.8 |
> |      w.r.t. $\alpha$      |                          |      |                 |              |        |          |            |      |
> |     $R_{(.)}$/0.5/0      |           4.54           | 39.7 |      37.8       |     62.9     |  8.2  |   8.2    |    57.1     | 33.4 |
> |       $R_{(.)}$/0/0       |           4.46           | 39.9 |      27.3       |      60.0      |  8.1   |   7.9    |    60.1    | 32.6 |
> |      w.r.t. $\beta$       |                          |      |                 |              |        |          |            |      |
> |         1/0.5/0          |           4.65           | 41.1 |      41.4       |     60.1     |  8.6  |   8.4   |    57.1    | 34.7 |
> |         1/0.25/0         |           4.59           | 38.8 |      40.3       |     61.5     |  8.3   |  7.9   |    60.6    | 34.3 |
> |         1/0.2/0          |           4.64           | 40.1 |      38.3       |     61.7     | 8.4  |   8.3    |    59.9    | 34.2 |
>
> From the table, it can be observed that:
>
> (i) $\gamma$ should be set to 0, as a non-zero $\gamma$ leads to a decrease in results.
>
> (ii) $\alpha$ should be set to 1, as setting $\alpha$ to the reward score results in a decrease in performance.
>
> (iii) The final results are insensitive to $\beta$, meaning that setting $\beta$ to a reasonable value will not significantly impact the overall performance.

---

> ### Author Response · Authors · 2023-11-17
> **3. External model requirement**
>
> **3. FIGA also requires an external model to be available, such as gpt-3.5-turbo. This is a huge limitation. Moreover, this should be ablated against other strategies that would also use gpt-3.5-turbo, such as distillation strategies.**
>
> (1) It is essential to recognize that employing an external model is a common practice in the construction of preference datasets. For instance, Anthropic HH-RLHF utilizes outputs from Anthropic's language models, and RRHF utilizes the outputs from ChatGPT to participate in the sorting of multiple responses. Such involvement of external models is almost ubiquitous in the creation of human preference datasets.
>
> (2) Moreover, utilizing gpt-3.5-turbo specifically offers a significant advantage. Considering **gpt-3.5-turbo's status as one of the most advanced language models available, it enables us to generate high-quality data, which is a feat not easily replicable by other models**. This approach has been validated in numerous studies, leading to the development of several impactful open-source models, like alpaca and vicuna.
>
> (3) For other methods that also use GPT-3.5-turbo, **our SFT mentioned in Table 2 is conducted on the revision of GPT-3.5-turbo, making it a form of distillation strategy**. The advantages of FIGA against distillation methods are evident from the comparisons in Table 2.
>
> (4) In addition, utilizing gpt-3.5-turbo to construct the SPA dataset does not take much of a budget, costing only $21.45 in total.

---

> ### Author Response · Authors · 2023-11-17
> **4. Other baselines**
>
> **4. More generally, the experiments lack many important baselines, in particular DPO [Rafailov2023]. I list other below, perhaps less critical and more recent, that adding them would be a big positive point.**
>
> Thanks for your valuable suggestion. We have added **DPO, ReST, SliC-HF, and RSO to the baseline methods** that we compare with.
> In considering the implementation of the three baseline methods—ReST, SliC-HF, and RSO—it's important to note that these methods were originally designed to operate independently of external models like ChatGPT. This consideration is crucial for maintaining a fair comparative basis among the baselines. Therefore, we follow the setting in the original paper and continue our experiments on the SFT model for them rather than directly training on the Alpaca model.
>
> Additionally, given that we already have an existing reward model, we opt for a specific sampling method for both SLiC-HF and RSO. For each input, we generate eight responses and select the two with the highest and lowest rewards for training. The key difference between SLiC-HF and RSO lies in their loss functions, which are hinge and hinge-norm respectively.
> Other parameter settings follow the approach outlined in Liu2023b.
>
> It can be seen that **FIGA also outperforms others with a large margin, and still stands out as the best one**.
>
> | Methods             | Reward | MMLU | TruthfulQA | Crows-Pairs | WinoGender | Vicuna | WizardLM | Average | Average_new |
> | ------------------- | ------ | ---- | ---------- | ----------- | ---------- | ------ | -------- | ------- | ----------- |
> | Alpaca-7b | 3.96   | 39.2 | 33.7       | 61.1        | 55.6       | 7.9    | 7.0        | 31.7    | 50.9        |
> | DPO  | 4.23   | 40.1 | 34.8       | 61.2        | 57.0         | 8.0      | 7.7      | 32.7    | 52.8        |
> | ReST | 4.52   | 39.1 | 23.6       | 61.8        | 55.4       | 8.2    | 8.2      | 31.1    | 52.2        |
> | SLiC-HF   | 4.57   | 39.6 | 30.1       | 60.7        | 57.6       | 8.4    | 8.3      | 32.7    | 54.2        |
> | RSO     | 4.58   | 39.4 | 31.8       | 60.1        | 58.0         | 8.5    | 8.4      | 33.1    | 54.8        |
> | FIGA   | 4.61   | 41.0   | 37.0         | 59.6        | 58.1       | 8.5    | 8.3      | 34.2    | 55.8        |

---

> ### Author Response · Authors · 2023-11-17
> **5. Minors**
>
> Thank you for pointing out the typo in the legend of Figure 1, and we have corrected it. Regarding the transformation of the SPA dataset construction into an online format and experiments using a larger parameter scale, we are actively conducting research and plan to include the results in a future version.

---

> ### Author Response · Authors · 2023-11-17
> **6. Question1: LLM's identification of high/low-quality part**
>
> **5. Rather than using Levenshtein distance, I am interested in knowing whether you could directly ask gpt-3.5-turbo to explicitly state which part of the sentence is correct/false.**
>
> Actually, in our paper, we have explored various variants of Levenshtein distance to identify which part of the sentence is correct/false. Specifically, **we have tried two methods using GPT-3.5-turbo and a method based on bag of words**, with experimental results shown in Table 6.
>
> From the results, it can be observed that Levenshtein distance is the most effective in leveraging the valuable information within sentences for our approach.

---

> ### Author Response · Authors · 2023-11-17
> **7. Question2: Details of generating revision using gpt-3.5-turbo**
>
> **6. Moreover, to further improve results, that would be interesting to give to gpt-3.5-turbo both the human-preferred generation along with the agent generation. Somehow, gpt-3.5-turbo could keep high rewards while reducing the #ops.**
>
> Actually, this is how FIGA guides GPT-3.5-turbo in generating revisions: **we simultaneously provide GPT-3.5-turbo with both the human-preferred generation and the agent generation**, enabling it to maintain high rewards while reducing the number of operations. The specific details of the prompts we use are outlined in Appendix A.2.

---

> ### Author Response · Authors · 2023-11-20
> **Kindly Reminder for the Discussion**
>
> Dear Reviewer hMwD,
>
> Thanks for your careful reading of our paper. We have tried our best to elaborate on the unclear points and revised our paper accordingly.
>
> We have **added DPO, ReST, SliC-HF, and RSO as the baseline methods** to compare with our FIGA method. Great thanks for your reminder. In addition, we **refined our experiments about hyperparameters**, showing that our seemingly numerous hyperparameters are not sensitive in practice, and therefore do not become a limitation of our method. If there are any other experiments you believe are necessary to elucidate the results, we will make every effort to complete them.
>
> We would like to know whether you find our response satisfactory, or if there are more questions that we could clarify. Since the rebuttal stage is coming to an end, we are more than happy to hear your comments and address any of your further concerns during the remaining time.
>
> Best,
>
> Authors

---

> > ### Comment · Reviewer_hMwD · 2023-11-21
> > **Response**
> >
> > Based on the other reviews and the detailed answers, I will update my score to 6.

---

> > > ### Author Response · Authors · 2023-11-21
> > >
> > > Dear Reviewer hMwD,
> > >
> > > We sincerely thank you for the updated score.
> > >
> > > Thanks for your constructive review. We sincerely appreciate your time and effort in reviewing our paper and reading our comments. Your review really helped us greatly in improving our paper, and we are truly grateful for your comments.  We will add the baseline methods you suggest in our final version. If you also have any other questions, please feel free to let us know. We will continue to try our best to answer for you.
> > >
> > > Best,
> > >
> > > Authors

---

### Official Review · Reviewer_1jST · 2023-10-15

**Soundness:** 3 good
**Presentation:** 3 good
**Contribution:** 4 excellent
**Rating:** 8
**Confidence:** 3

**Summary:**

The authors present a method FIGA as a new alignment method with more fine-grained signal compared to e.g., RLHF. For a given text output task, they obtain pairs of poor responses $\hat{Y}$ and strong responses $Y$, and make local changes to $\hat{Y}$ based on qualities of $Y$ to obtain an improved response $\tilde{Y}$ that is still surface-level-similar to $\hat{Y}$. Comparing $\tilde{Y}$ to the original $\hat{Y}$ then gives us fine-grained signal on particular token positions which can be used for more targeted SFT. They show improvement on a few different benchmarks.

**Strengths:**

--interesting, novel method for generating pairs of responses with localized improvements for fine-grained feedback

--i can definitely see the main intuition (distilling the qualities of a strong response into a more surface-level-similar response) as potentially inspiring future work

**Weaknesses:**

--how do you collect the human-preferred response $Y$? (i.e., the one that you feed into ChatGPT together with $\hat{Y}$ to get $\tilde{Y}$). based on Fig1, i assume this is not generated by the same model as used to get $\hat{Y}$, but either provided with the dataset or generated by ChatGPT in a distillation-like setting? but since you have a threshold $\eta_2$ for filtering the quality of $Y$ anyway, i wonder if you could generate $Y$ using your model as well-- I think this would really strengthen your contribution, demonstrating that the method works even without supervision / distillation from a stronger model; your method would then operate in a setup closer to e.g., RLAIF where you only need the initial prompts and principles, and generate all outputs/evaluations yourself without accessing stronger model. you'd only need ChatGPT for the part where you turn $Y$ and $\hat{Y}$ into $\tilde{Y}$, which I think isn't really distillation in the same sense.

--seems like $\beta$ is set to 0, so you built up this whole framework but then in practice turned off a good chunk of it (the part where you penalize the initial bad response)? see questions below

**Questions:**

--on the topic of $\beta$ being set to 0, i saw that even in your later ablation you only set $\beta$ to 0.025. If you increase $\beta$, do you get the issue with unlikelihood training [1] where (I believe) it's been shown that you unnecessarily penalize unrelated tokens that are just fluent in context?

[1] Welleck, Sean, et al. "Neural text generation with unlikelihood training." arXiv preprint arXiv:1908.04319 (2019).

--similarly, in eqn 2, if e.g. hypothetically you make a change that touches several parts of the text in a consistent way (like say someone's name changes from Amy to Bob in 3 places), you might be penalizing the parts that come later even if they're simply being consistent with what was written elsewhere, which might throw off your optimization if you put nonzero weight on the second term? Not sure if this would help, but what might happen if you were to optimize by e.g., not using $\hat{Y}$ and $\tilde{Y}$ in their entirety, but doing more "atomic" changes like only changing one span at a time? Is it too difficult to align the spans?

--thoughts on using this kind of method for tasks requiring even longer outputs? since i wonder if this fine-grained feedback is more valuable in longer-context settings, so i wonder if you would get larger performance gains compared to baselines. though those tasks might be more challenging to evaluate.

--how consistent are the revisions from ChatGPT having smallish levenshtein distance between $\hat{Y}$ and $\tilde{Y}? any analysis (even qualitative) on how good ChatGPT's categorization of the 4 types of issues is, and how good are its revisions at addressing these? would be curious to see this analysis since it could point to potential future directions for improvement

(note- i suggested a few different things here. to be clear, i definitely don't expect further experiments on all or even any of these.)

---

> ### Author Response · Authors · 2023-11-17
> **1. Other ways to collect human-preferred responses**
>
> Thanks for your detailed and meaningful comments! We have listed our response to your concerns as follows. If you also have any other questions, please feel free to let us know. We will continue to try our best to answer for you.
>
> **1. how do you collect the human-preferred response Y? Based on Fig1, i assume this is not generated by the same model as used to get $\hat{Y}$, but either provided with the dataset or generated by ChatGPT in a distillation-like setting? but since you have a threshold for filtering the quality of Y anyway, i wonder if you could generate Y using your model as well-- I think this would really strengthen your contribution, demonstrating that the method works even without supervision / distillation from a stronger model; your method would then operate in a setup closer to e.g., RLAIF where you only need the initial prompts and principles, and generate all outputs/evaluations yourself without accessing stronger model.**
>
> Thanks for your insightful idea, it greatly excited us when we first encountered it. We conduct some preliminary experiments, such as guiding the Alpaca-7B model to generate a better response based on reflection, to replace our current use of human-preferred responses.
>
> However, the practical challenge we encounter is that even under the reflection setting, Alpaca-7B struggles to produce outputs of comparable quality to ground truth. This difficulty may arise from the motivation behind the FIGA method, which aims to focus on queries where the initial model performs poorly. Therefore, for now, using ground truth as guidance for revision proves to be more stable.

---

> ### Author Response · Authors · 2023-11-17
> **2. Penalize unrelated tokens**
>
> **2. on the topic of $\beta$ being set to 0, i saw that even in your later ablation you only set $\beta$ to 0.025. If you increase $\beta$, do you get the issue with unlikelihood training [1] where (I believe) it's been shown that you unnecessarily penalize unrelated tokens that are just fluent in context?**
>
> We increase the weight of the penalty term $\beta$ to 0.5, which is **FIGA_new** in the table below.
>
> | $\alpha / \beta / \gamma$ | Reward | MMLU | TruthfulQA | Crows-Pairs | WinoGender | Vicuna | WizardLM | Average |
> | ------------------------- | ------ | ---- | ---------- | ----------- | ---------- | ------ | -------- | ------- |
> | Alpaca-7b                 | 3.96   | 39.2 | 33.7       | 61.1        | 55.6       | 7.9    | 7.0        | 31.7    |
> | FIGA (1/0/0)              | 4.61   | 41.0   | 37.0         | 59.6        | 58.1       | 8.5    | 8.3      | 34.2    |
> | FIGA_new (1/0.5/0)        | 4.62   | 40.8 | 42.0         | 61.2        | 59.6       | 8.6    | 8.3      | 34.9    |
>
> It can be observed that penalizing words that harm the quality led to an overall improvement in performance. We do not encounter the issue with unlikelihood training, and the reasons could be:
>
> (i) The construction of the SPA dataset ensures that the penalized words filtered out are indeed poor-quality, degrading words.
>
> (ii) In practical implementation, when we increase the weight of the penalty term $\beta$ to 0.5, we further select the bad tokens returned by the Levenshtein distance by retaining only those with a negative log-likelihood less than 0.6. In this way, the number of tokens we penalize in each sentence is around 5.

---

> ### Author Response · Authors · 2023-11-17
> **3. FIGA for tasks requiring longer outputs**
>
> **3. Thoughts on using this kind of method for tasks requiring even longer outputs? Since i wonder if this fine-grained feedback is more valuable in longer-context settings, so i wonder if you would get larger performance gains compared to baselines. Though those tasks might be more challenging to evaluate.**
>
> To explore this scenario, we conduct a detailed analysis of our method's performance across different output lengths. We divide our test set based on the output lengths of FIGA's generation (0-500, 500-1000, 1000+ characters) and compare the reward scores before and after FIGA training. Our findings are summarized in the table below:
>
> | Output Length Category | Reward Score Before FIGA Training | Reward Score After FIGA Training |
> | ---------------------- | --------------------------------- | -------------------------------- |
> | 0-500                  | 2.78                              | 3.85                             |
> | 500-1000               | 4.84                              | 5.17                             |
> | >1000                  | 5.52                              | 5.79                             |
>
> These results demonstrate consistent performance improvements across all categories, with significant gains in both shorter and longer output tasks. While the relative gains in longer output lengths appear to be somewhat reduced compared to shorter outputs, they are still notable and affirm the effectiveness of our method. This finding highlights the versatility of our method, suggesting its efficacy even in complex scenarios requiring extended responses.
>
> Furthermore, while the current analysis offers initial insights, we acknowledge its limitations and are excited about the potential of our method in broader applications. We are particularly interested in exploring its efficacy in tasks with longer inputs as well as outputs as a part of our future work. Thanks for your insightful question.

---

> ### Author Response · Authors · 2023-11-17
> **4. Consistency of ChatGPT's revisions about Levenshtein distance**
>
> **4. How consistent are the revisions from ChatGPT having smallish levenshtein distance between $\hat{Y}$ and $\tilde{Y}$?**
>
> As highlighted in your question, this aspect forms a crucial premise in constructing our SPA dataset. We affirm that ChatGPT is indeed proficient in this specific task of making minimal yet effective modifications to enhance the quality of responses.
>
> (i) Our findings, as detailed in Table 1 and Figure 2 of our paper, provide evidence supporting this claim. We observed a significant decrease in the average number of token-level modifications per sentence from 75.69 in the original response to 39.38 post-ChatGPT revision. Concurrently, there was a notable increase in the reward score, from 1.78 to 3.94. These metrics indicate ChatGPT's ability to enhance response quality while ensuring minimal deviation, as measured by the Levenshtein distance.
>
> (ii) To directly illustrate ChatGPT's revision effectiveness, we selected an example with a modification count close to the average (40).
>
> **Instruction:** With her savings, Sara bought two books: a book for 5.5£ and a book for 6.5£. She gives a 20£ bill to the seller. How much change does she get back? Give me reasons, before answering the question
>
> ​	**$Y$:** Sara will get back 10£ in change. This is because the price of the cheaper book was 5.5£, so Sara paid 11.5£ and the seller will give her back the 10£ difference. Similarly, the price of the more expensive book was 6.5£, so Sara paid 12.5£ and the seller will give her back the 10£ difference.
>
> ​	**$\hat{Y}$:** To calculate the change Sara will get back, we need to first find out how much she spent on the books.\n\nCost of the first book: 5.5£\nCost of the second book: 6.5£\n\nTotal cost of the books: 5.5£ + 6.5£ = 12£\n\nSara gave the seller a 20£ bill.\n\nChange = Amount paid - Total cost of the books\nChange = 20£ - 12£ = 8£\n\nSo, Sara will get 8£ back in change. This is because she spent a total of 12£ on the books and paid with a 20£ bill. The difference between the cost of the books and the bill she paid with is her change.
>
> ​	**$\tilde{Y}$:** Sara will get back 8£ in change. This is because she gave the seller a 20£ bill, and the total cost of the books she bought was 12£. The difference between the amount she paid and the total cost of the books is her change.
>
> It is evident that ChatGPT maintains the structural and stylistic elements of the original response (Y), such as the phrases "Sara will get back xx£ in change" and "This is because...". Achieving this level of modification, akin to sentence rewriting, is indeed challenging. The task demands a balance between accuracy and adherence to the original style, illustrating the complexity involved in such revisions. We look forward to further research in this area to enhance and refine these capabilities.

---

> ### Author Response · Authors · 2023-11-17
> **5. Analysis of ChatGPT's categorization of the 4 types of issues and the quality of its revisions**
>
> **5. Any analysis (even qualitative) on how good ChatGPT's categorization of the 4 types of issues is, and how good are its revisions at addressing these? Would be curious to see this analysis since it could point to potential future directions for improvement.**
>
> (i) Categorization Accuracy:
>
> In our original research design, we included a test to evaluate ChatGPT's accuracy in categorizing four distinct types of issues, comparing its performance against that of one human adjudicator. This test was integral to our methodology, consisting of 48 questions divided equally among the four error types. ChatGPT achieved a notable accuracy rate of 77% in aligning with the human adjudicators' categorizations. This pre-planned assessment of ChatGPT's categorization capability was a crucial part of our study, affirming our confidence in employing ChatGPT for accurately identifying and categorizing these issue types.
>
> The specific results are illustrated in the table below:
>
> | Error Type | Description                                                  | ChatGPT's Accuracy Rate |
> | ---------- | ------------------------------------------------------------ | ----------------------- |
> | A          | Needs more accurate content                                  | 75%                     |
> | B          | Needs more comprehensive content or more details             | 75%                     |
> | C          | Requires adjustments in structure                            | 67%                     |
> | D          | Other reasons (such as containing harmful information or going off-topic) | 92%                     |
>
> (ii) Quality of Revisions:
>
> As for the quality of ChatGPT's revisions, this was addressed in a previous answer, where we highlighted ChatGPT's commendable ability to follow instructions and implement changes effectively.

---

> ### Author Response · Authors · 2023-11-20
> **Kindly Reminder for the Discussion**
>
> Dear Reviewer 1jST,
>
> Thanks for your careful reading of our paper. We have tried our best to elaborate on the unclear points and revised our paper accordingly.
>
> Specifically, we have **conducted experiments with a greater $\beta$**, and evaluate FIGA's performance for tasks that require longer outputs. In addition, we carried out a **human evaluation** of how ChatGPT performs on error classification and revision. Besides, we carefully explained your thoughts about other ways to collect human-preferred responses. If there are any other experiments you believe are necessary to elucidate the results, we will make every effort to complete them.
>
> We would like to know whether you find our response satisfactory, or if there are more questions that we could clarify. Since the rebuttal stage is coming to an end, we are more than happy to hear your comments and address any of your further concerns during the remaining time.
>
> Best,
>
> Authors

---

> > ### Comment · Reviewer_1jST · 2023-11-21
> > **Thanks for the response**
> >
> > Thank you for running the new experiments and for the interesting discussions. I will maintain my score of 8.
> >
> > Btw, I think it's totally fine if the RLAIF-style setting doesn't quite work in the current version (re: "Other ways to collect human-preferred responses"), and it's definitely 100% expected that it won't work quite as well as the version using ground-truth human responses. I think it'd still be interesting to see that discussed in an appendix if it's not too much trouble to add (just showing whatever you have from your preliminary analysis that you mentioned, no need to do extra work), even if the initial results are negative; maybe just with some discussion/speculation for what could be changed to get it to work better.

---

> > > ### Author Response · Authors · 2023-11-21
> > >
> > > Dear Reviewer 1jST,
> > >
> > > Thanks for your constructive review. We sincerely appreciate your time and effort in reviewing our paper and reading our comments. And we will provide the results and analysis for RLAIF-style setting in the appendix of the final version. Great thanks for the valuable advice and discussion. Your review really helped us greatly in improving our paper, and we are truly grateful for your response.
> > >
> > > Best,
> > >
> > > Authors

---

### Official Review · Reviewer_772X · 2023-10-28

**Soundness:** 2 fair
**Presentation:** 4 excellent
**Contribution:** 2 fair
**Rating:** 6
**Confidence:** 5

**Summary:**

The paper proposes FIGA, an RL-less method that aligns LLMs with fine-grained token-level quality signals. A new curated dataset SubPar Alignment (SPA) is constructed as well, which contains pairs of initial/revised responses. FIGA is evaluated on a diverse set of benchmark tasks, outperforming baselines including SFT and PPO.

**Strengths:**

- The paper is well written and is easy to follow.
- The motivation of the paper is timely and important.
- Many in-depth analysis on the constructed dataset and different experiment settings.

**Weaknesses:**

- The paper needs more comparison with works that do not leverage reward models (e.g. DPO [1]), to demonstrate the advantage over recent RLless methods
- The paper needs to specify what each performance metric is for the benchmark datasets.
- In Table 2, performance improvement is marginal compared to the best performing baseline method. Also, multiplying the Reward metric by 10 while leaving Vicuna, WizardLM at the original scale is misleading.
- In Figure 3, the win rate of FIGA is below 50% most of the time, questioning the actual advantage of FIGA over baselines. I do not think that being a  “Tie” justifies the necessity of FIGA sufficiently.
- There are quite a number of hyperparameters to tune. How sensitive is the method to alpha, beta and gamma?

[1] Direct Preference Optimization: Your Language Model is Secretly a Reward Model (R.  Rafailov et al., 2023)

**Questions:**

See Weaknesses

---

> ### Author Response · Authors · 2023-11-17
> **1. Comparison with other recent RL less methods**
>
> Thanks for your detailed and meaningful comments! We have listed our response to your concerns as follows. If you also have any other questions, please feel free to let us know. We will continue to try our best to answer for you.
>
> **1. The paper needs more comparison with works that do not leverage reward models (e.g. DPO [1]), to demonstrate the advantage over recent RL less methods.**
>
> We have expanded our baseline selection to include DPO, and have conducted a series of benchmarks to evaluate its performance. The detailed results of these benchmarks can be found in Table 2 both in our updated paper and below.
>
> | Methods   | Reward | MMLU | TruthfulQA | Crows-Pairs | WinoGender | Vicuna | WizardLM | Average |
> | --------- | ------ | ---- | ---------- | ----------- | ---------- | ------ | -------- | ------- |
> | Alpaca-7b | 3.96   | 39.2 | 33.7       | 61.1        | 55.6       | 7.9    | 7.0        | 31.7    |
> | DPO       | 4.23   | 40.1 | 34.8       | 61.2        | 57.0         | 8.0      | 7.7      | 32.7    |
> | FIGA      | 4.61   | 41.0   | 37.0         | 59.6        | 58.1       | 8.5    | 8.3      | 34.2    |
>
> It can be seen that **FIGA also outperforms DPO with a large margin, and still stands out as the best one.**
>
> Additionally, we are in the process of incorporating other baselines suggested by other reviewers. Please stay tuned for our further experimental results.

---

> ### Author Response · Authors · 2023-11-17
> **2. Performance metrics for the benchmark datasets**
>
> **2. The paper needs to specify what each performance metric is for the benchmark datasets.**
>
> Thanks for your suggestion, we have added detailed explanations of each performance metric for the benchmark datasets in Appendix A.3 of our paper.

---

> ### Author Response · Authors · 2023-11-17
> **3. Performance Improvement**
>
> **3. In Table 2, performance improvement is marginal compared to the best performing baseline method. Also, multiplying the Reward metric by 10 while leaving Vicuna, WizardLM at the original scale is misleading.**
>
> (1) The results in Table 2 demonstrate that FIGA performs significantly better compared to the baselines. Among all baseline methods training on the same training set as ours, FIGA achieves the best performance, with the average benchmark metric exceeding at least 1.5.
>
> Notably, **FIGA outperforms PPO (i.e. PPO 85k mentioned in Table 2) training on the dataset which utilizes over four times the amount of SPA by 1.1 points**. PPO is currently one of the most mainstream alignment methods and this variant utilizes much more data than we use. Despite this, its effectiveness lags behind ours, providing sufficient evidence for the efficacy of FIGA.
>
> (we have again attached Table 2 in our paper below for your convenience, and comment on the size of each training dataset)
>
> | Methods             | Reward   | MMLU     | TruthfulQA | Crows-Pairs | WinoGender | Vicuna  | WizardLM | Average  | Average_new |
> | :------------------ | :------- | :------- | ---------- | ----------- | ---------- | ------- | -------- | -------- | ----------- |
> | Alpaca-7b | 3.96     | 39.2     | 33.7       | 61.1        | 55.6       | 7.9     | 7.0      | 31.7     | 50.9        |
> | SFT (SPA 17k)       | 4.56     | 39.3     | 22.0       | 61.5        | 55.3       | 8.4     | 8.3      | 31.1     | 52.5        |
> | PPO (SPA 17k)       | 4.06     | 39.6     | 30.1       | 61.3        | 56.2       | 7.6     | 7.4      | 31.5     | 50.7        |
> | PPO (85k)           | 4.54     | 39.2     | 36.7       | 60.6        | 56.2       | 7.9     | 7.2      | 33.1     | 52.6        |
> | CoH (SPA 17k)       | 4.24     | 39.6     | 28.2       | 59.6        | 52.1       | 8.3     | 8.1      | 32.7     | 52.4        |
> | RRHF (SPA 17k)      | 4.23     | 37.8     | 32.9       | 59.9        | 60.0       | 7.9     | 7.9      | 31.3     | 53.0        |
> | DPO (SPA 17k)   | 4.23 | 40.1 | 34.8   | 61.2    | 57.0    | 8.0   | 7.7  | 32.7 | 52.8   |
> | **FIGA (SPA 17k)**  | **4.61** | **41.0** | **37.0**   | **59.6**    | **58.1**   | **8.5** | **8.3**  | **34.2** | **55.8**    |
>
> (2) As for the Average metric, we use it to comprehensively consider all the benchmarks selected, reflecting the model's overall performance.
>
> Considering that both Vicuna and WizardLM of FIGA outperform all other baselines, and the disparities between different baselines are already relatively large in these two benchmarks even with a total score of only ten, and after multiplying by ten, FIGA's lead in these two benchmarks is significantly greater than in other benchmarks, which makes it not beneficial to consider the model's performance across multiple benchmarks. Therefore, in the paper, we did not multiply them by ten.
>
> However, we also computed the **Average_new** by multiplying the rewards, Vicuna, and WizardLM by ten. It can be observed that FIGA's advantage is further amplified.
>
> May I seek your esteemed opinion on whether it would be preferable for us to assess the overall model performance using "Average_new" or "Average" in the final analysis?

---

> ### Author Response · Authors · 2023-11-17
> **4. Win Rate of FIGA**
>
> **4. In Figure 3, the win rate of FIGA is below 50% most of the time, questioning the actual advantage of FIGA over baselines. I do not think that being a “Tie” justifies the necessity of FIGA sufficiently.**
>
> (1) In Figure 3, many instances of ties occur because our evaluation process is:
>
> ​	(i) Initially having ChatGPT score the responses,
>
> ​	(ii) and then determining the outcome as a win/lose/tie based on the scores.
>
> Ties may arise when two responses have the same score, although their quality might have subtle differences.
>
> In the current scenario, considering that **FIGA wins more often than all baseline methods** except for ChatGPT, it can be inferred that FIGA demonstrates superiority.
>
> (2) To prevent the aforementioned ties resulting from equal scores between two responses, we **directly** let ChatGPT make a pairwise comparison between the responses from FIGA and another baseline, and select the one of higher quality. The results are as follows.
>
> | FIGA against  | Alpaca-7b | SFT  | PPO 17k | PPO 85k | RRHF | CoH  | ChatGPT |
> | ------------- | ------ | ---- | ------- | ------- | ---- | ---- | ------- |
> | FIGA win rate | 79%    | 61%  | 83%     | 81%     | 76%  | 60%  | 33%     |
>
> (3) Additionally, we introduced **human evaluations**, providing a more comprehensive test of the effectiveness of FIGA.
>
> We randomly selected 30 instructions from the Vicuna benchmark, along with corresponding responses from FIGA and other baseline methods. We invited three evaluators to make pairwise comparisons between FIGA responses and those of each baseline method, aiming to identify the one with higher quality. Finally, we aggregated the assessments from the three participants and obtained the final human evaluation results for these 30 instructions through the mode method.
>
> | FIGA against  | Alpaca-7b | SFT  | PPO 17k | PPO 85k | RRHF | CoH  | ChatGPT |
> | ------------- | ------ | ---- | ------- | ------- | ---- | ---- | ------- |
> | FIGA win rate | 73%    | 57%  | 73%     | 83%     | 73%  | 57%  | 40%     |
>
> We can see from the results that FIGA outperforms all baseline methods, indicating its ability to better align with human preferences.

---

> ### Author Response · Authors · 2023-11-17
> **5. Sensitivity of hyper-parameters**
>
> **5. There are quite a number of hyperparameters to tune. How sensitive is the method to alpha, beta and gamma?**
>
> (1) In reality, we don't have a multitude of parameters to tune. The current hyperparameters are included for the sake of method completeness and clear exposition. Setting $\alpha=1$ and $\gamma=0$ is a natural choice since we need to emphasize positive words and ignore those that would naturally occur.
>
> (2) To further illustrate, we conducted experiments demonstrating that these **hyperparameters are not sensitive**.
>
> | $\alpha / \beta / \gamma$ | Reward | MMLU | TruthfulQA | CrowS-Pairs | Vicuna | WizardLM | WinoGender | Average  |
> | :-----------------------: | :----------------------: | :--: | :-------------: | :----------: | :----: | :------: | :--------: | :--: |
> |       (FIGA) 1/0/0        |           4.61           |  41.0  |       37.0        |     59.6     |  8.5   |  8.3   |    58.1    | 34.2 |
> |      w.r.t. $\gamma$      |                          |      |                 |              |        |          |            |      |
> |        1/0.5/0.3         |           4.54           | 41.2 |      32.2       |     60.1     |  8.4   |   8.2    |     56.0     |  33.0  |
> |          1/0/0.3          |           4.50            | 40.4 |      30.8       |     59.8     |  8.4   |   8.1    |    56.4    | 32.8 |
> |      w.r.t. $\alpha$      |                          |      |                 |              |        |          |            |      |
> |     $R_{(.)}$/0.5/0      |           4.54           | 39.7 |      37.8       |     62.9     |  8.2  |   8.2    |   57.1     | 33.4 |
> |       $R_{(.)}$/0/0       |           4.46           | 39.9 |      27.3       |      60.0      |  8.1   |   7.9    |    60.1    | 32.6 |
> |      w.r.t. $\beta$       |                          |      |                 |              |        |          |            |      |
> |         1/0.5/0          |           4.65           | 41.1 |      41.4       |     60.1     |  8.6  |   8.4   |    57.1    | 34.7 |
> |         1/0.25/0         |           4.59           | 38.8 |      40.3       |     61.5     |  8.3   |  7.9   |    60.6    | 34.3 |
> |         1/0.2/0          |           4.64           | 40.1 |      38.3       |     61.7     | 8.4  |   8.3    |    59.9    | 34.2 |
>
> From the table, it can be observed that:
>
> (i) $\gamma$ should be set to 0, as a non-zero $\gamma$ leads to a decrease in results.
>
> (ii) $\alpha$ should be set to 1, as setting $\alpha$ to the reward score results in a decrease in performance.
>
> (iii) The final results are insensitive to $\beta$, meaning that setting $\beta$ to a reasonable value will not significantly impact the overall performance.

---

> ### Author Response · Authors · 2023-11-20
> **Kindly Reminder for the Discussion**
>
> Dear Reviewer 772X,
>
> Thanks for your careful reading of our paper. We have tried our best to elaborate on the unclear points and revised our paper accordingly.
>
> We have **added DPO as a baseline method** to compare with our FIGA method. In addition, we follow your advice to **add performance metrics for each benchmark dataset to Appendix A.3** in our revised paper. Besides, we further discuss the performance of FIGA to explain that it performs pretty well, and we also conduct a **human evaluation** to better explain this. Also, we refined our experiments about hyperparameters, showing that **our seemingly numerous hyperparameters are not sensitive in practice**, and therefore do not become a limitation of our method. If there are any other experiments you believe are necessary to elucidate the results, we will make every effort to complete them.
>
> We would like to know whether you find our response satisfactory, or if there are more questions that we could clarify. Since the rebuttal stage is coming to an end, we are more than happy to hear your comments and address any of your further concerns during the remaining time.
>
> Best,
>
> Authors

---

> ### Comment · Reviewer_772X · 2023-11-20
> **Reviewer Response**
>
> Dear Authors,
>
> Thank you very much for your responses. I appreciate the extensive experiments the authors have further provided. My major concerns have been resolved, and I am raising my score.
>
> As for question 3, I suggest the authors present the average value as they wish. But providing the rationale behind the way they computed the average value will be helpful.
>
> Thank you,
>
> Reviewer 772X

---

> > ### Author Response · Authors · 2023-11-20
> >
> > Dear Reviewer 772X,
> >
> > We sincerely thank you for the updated score.
> >
> > Thanks for your constructive review. We sincerely appreciate your time and effort in reviewing our paper and reading our comments. And we will provide the rationale behind the way we compute the average value in the final version. Your review really helped us greatly in improving our paper, and we are truly grateful for your comments. If you also have any other questions, please feel free to let us know. We will continue to try our best to answer for you.
> >
> > Best,
> >
> > Authors

---

### Official Review · Reviewer_J3kJ · 2023-11-01

**Soundness:** 3 good
**Presentation:** 3 good
**Contribution:** 3 good
**Rating:** 6
**Confidence:** 3

**Summary:**

This paper mainly tackles the problem of insufficient supervision when aligning LLM's responses to human-preferred ones. The authors argue that the current paradigm of only giving the developed model ("rollout model") the quality label (e.g., good or bad) of the demonstration and instructing the rollout model to imitate human-preferred responses is insufficient, as the rollout model may not fully capture the correct alignment behaviors. Therefore, this training paradigm can make the rollout model suffer from the distribution shift between its own outputs and imitated outputs.

The author proposes to bridge the gap of this distribution mismatch by creating a surrogate target that is more similar (tries to maintain a small edit distance) to the model's original low-quality outputs, but is of better quality. The creation of such a target is mainly done by first using a pretrained reward model to filter out queries where the model gives low-quality output, and then instructing ChatGPT, the state-of-the-art LLM, to compare the developed model's original outputs and the human-preferred and do minimum edits to improve the original outputs.

The authors build a new dataset SPA based on previous human preference datasets by replacing the human demonstration with the surrogate target and training the developed model to edit its outputs to surrogate targets via a token-wise loss. Experiment results demonstrate the effectiveness of the proposed method over various tasks.

**Strengths:**

1. This paper is well-motivated, makes a timely contribution to the distribution shift problem in the Human-LLM Alignment field, and contributes a useful LLM-in-the-loop method to create a higher-quality dataset better for model fine-tuning.

2. The experiment results are comprehensive and solid enough to support the major claims of the paper.

**Weaknesses:**

1. **Narrowed Application Range**: It seems the proposed dataset creation method is strongly biased by the rollout model checkpoint and the reward model used for identifying low-quality outputs. As each rollout model and reward model combination may have different failure modes, it is not clear if the practitioners want to switch to other rollout models and other reward models, how much this dataset can still be helpful. It can be an unreasonably large consumption of computation power if we need to re-apply the proposed method for every rollout model and reward model checkpoint. This might be the problem for almost all human preference datasets as they also use rollout outputs from some model (e.g., Anthropic HH-RLHF uses Anthropic's own LLM outputs), but there seems to be more model-specific limitations for the dataset created by this method as the token-wise label used in the new dataset is also model-dependent. It would be better if the author could discuss about how much data is necessary to achieve reasonable performance gain, whether it can be applied to other LLMs (not only Alpaca-7B), and the budget decomposition.

2. **Possible Unfair Comparison to Baseline**: The reviewer noticed that in Section 4.2 and Section 4.3.1, the dataset used for the proposed training method and baseline methods are not of the same size. This may or may not give an unfair advantage to the proposed method. It would be better if the author could verify the effectiveness of the proposed method by subsampling the created dataset to double-check the effect of the dataset size.

3. **Unclear Alignment to Human Preference, and Insufficient Evaluation**:  The goal of Human-LLM Alignment is to make the model outputs more aligned with human values. It is okay to use a surrogate objective to better train the model for optimization purposes. However, it is important to ensure the training is still making progress towards the original goal. Therefore, there should be some evaluation on whether the model trained in the newly proposed way is getting closer to human preference.  A straightforward yet missing "inherent" evaluation is comparing the updated rollout model outputs to the original human-preferred ones -- how similar it is, and what is the proportion of samples getting more similar. Also, after training, whether the model truly learn to do "desired behaviors"? Has the model learned to avoid any of the aforementioned four types of mistakes? (In Section 3.1) The specific Reward Model can be biased and only reading "Reward" numbers at tables hardly gives readers confidence in the alignment performance in terms of human preference. This also applies to other benchmark numbers -- what the model is learning under the proposed method? The current version of the paper lacks analysis on this side.

      As human preference is more complicated than the benchmarks listed in this paper, more careful investigation should be conducted on the user side to understand how such target-changing would influence user perception.  It has been widely reported that GPT-4-based Evaluation (e.g., Vicuna Benchmark) has various biases (e.g., positional bias, verbosity bias, and self-enhancement bias as reported in [1,2,3]) and cannot fully represent humans, so it would be helpful to do the human evaluation to examine the quality of the rollout model outputs after going through the proposed training method.

4. **Unclear Experiment Detail**: Some hyperparameter setups in Section 4.3 seem arbitrary (e.g., in FIGA-contra, FIGA-reward and FIGA-fix). Why does the author set up the hyperparameter like that?



References:

[1] Zheng, Lianmin, et al. "Judging LLM-as-a-judge with MT-Bench and Chatbot Arena." NeurIPS 2023.

[2] Wang, Peiyi, et al. "Large language models are not fair evaluators." arXiv preprint arXiv:2305.17926 (2023).

[3] Pezeshkpour, Pouya, and Estevam Hruschka. "Large language models sensitivity to the order of options in multiple-choice questions." arXiv preprint arXiv:2308.11483 (2023).

**Questions:**

It seems weird that using "Average" scores to compare the models trained under different methods -- the value magnitude of "Vicuna" and "WizardLM" ("Reward" scores are multiplied by 10 as the author noted in the footnote)  are not consistent with performance numbers in other benchmarks, and the test data numbers across different tasks are also not the same. What is the motivation for using this metric?

---

> ### Author Response · Authors · 2023-11-17
> **1. Narrowed Application Range**
>
> Thanks for your detailed and meaningful comments! We have listed our response to your concerns as follows. If you also have any other questions, please feel free to let us know. We will continue to try our best to answer for you.
>
> **1. It would be better if the author could discuss about how much data is necessary to achieve reasonable performance gain, whether it can be applied to other LLMs (not only Alpaca-7B), and the budget decomposition.**
>
> (1) When applying FIGA to other large language models (other than Alpaca-7B), given that our motivation is to **selectively modify the model's weaknesses**, a targeted SPA dataset still needs to be created.
>
> However, employing a new rollout model doesn't require experimenting with all possible reward models, because the reward model we utilize is the most commonly used one, and has high accuracy in evaluating response quality as shown in the LLaMA-2 paper. Besides, the reward model is employed during SPA dataset construction merely to filter out the query where the rollout model performance is subpar, thus the reward model we use will still be helpful when adapting to other language models.
>
> (2) Moreover, the cost of constructing the SPA dataset is not significant. We have conducted a budget estimation based on the usage of the GPT-3.5-turbo API for our SPA dataset. Considering the pricing of 0.0010 per 1K tokens and 0.0020 per 1K tokens, we calculated that the average input token count per data entry in the SPA dataset is 952, and the average output token count is 143. This leads to a total estimated cost of $21.45 for the entire SPA dataset, which is not a high expenditure.
>
> (3) Additionally, **FIGA can demonstrate performance improvement with a small amount of data**. Regarding the question "How much data is necessary to achieve reasonable performance gains", we conduct experiments by varying the size of the SPA dataset. The results below indicate that constructing a SPA dataset with 8000 data points and training based on the FIGA method leads to a substantial performance boost.
>
> | #data  | Reward | MMLU | TruthfulQA | CrowS-Pairs | Vicuna | WizardLM | WinoGender | Average  |
> | ------ | ------ | ---- | ---------- | ----------- | ------ | -------- | ---------- | ---- |
> | Alpaca | 3.957  | 39.2 | 33.7       | 61.1        | 7.9 | 7.0    | 55.6       | 31.7 |
> | 17k    | 4.61   | 41.0   | 37.0         | 59.6        | 8.5    | 8.3    | 58.1       | 34.2 |
> | 8k     | 4.58   | 41.0   | 33.2       | 59.8        | 8.4    | 8.2     | 57.2       | 33.4 |
> | 4k     | 4.28   | 39.9 | 26.3       | 59.0          | 8.6    | 8.4      | 55.7       | 31.8 |

---

> ### Author Response · Authors · 2023-11-17
> **2. Possible Unfair Comparison to Baseline**
>
> **2. The reviewer noticed that in Section 4.2 and Section 4.3.1, the dataset used for the proposed training method and baseline methods are not of the same size. This may or may not give an unfair advantage to the proposed method. It would be better if the author could verify the effectiveness of the proposed method by subsampling the created dataset to double-check the effect of the dataset size.**
>
> As for your concern in Section 4.2, we apologize for any misunderstandings that may have arisen from our paper, but we have to state that our comparison is fair. We have reattached Table 2 in our paper below and have commented on the size of each training dataset. **It's important to note that for different baselines, we employed the same SPA dataset for training, so our comparison is fair**.
>
> Among them, PPO (85K) is the only one exception that does not use SPA dataset. The motivation of introducing this variant is that, we suppose for unsupervised methods like PPO, which do not utilize the response of each data entry, it is necessary to somewhat expand the dataset size to better leverage the potential of PPO.
>
> | Methods        | Reward | MMLU | TruthfulQA | Crows-Pairs | WinoGender | Vicuna | WizardLM | Average |
> | -------------- | ------ | ---- | ---------- | ----------- | ---------- | ------ | -------- | ------- |
> | Alpaca-7b      | 3.96   | 39.2 | 33.7       | 61.1        | 55.6       | 7.9    | 7.0        | 31.7    |
> | SFT (SPA 17k)  | 4.56   | 39.3 | 22.0         | 61.5        | 55.3       | 8.4    | 8.3      | 31.1    |
> | PPO (SPA 17k)  | 4.06   | 39.6 | 30.1       | 61.3        | 56.2       | 7.6    | 7.4      | 31.5    |
> | PPO (85k)      | 4.54   | 39.2 | 36.7       | 60.6        | 56.2       | 7.9    | 7.2      | 33.1    |
> | CoH (SPA 17k)  | 4.24   | 39.6 | 28.2       | 59.6        | 52.1       | 8.3    | 8.1      | 32.7    |
> | RRHF (SPA 17k) | 4.23   | 37.8 | 32.9       | 59.9        | 60.0         | 7.9    | 7.9      | 31.3    |
> | DPO (SPA 17k)  | 4.23   | 40.1 | 34.8       | 61.2        | 57.0         | 8.0      | 7.7      | 32.7    |
> | FIGA (SPA 17k) | 4.61   | 41.0   | 37.0         | 59.6        | 58.1       | 8.5    | 8.3      | 34.2    |
>
> Regarding your concern noted in Section 4.3.1, we want to clarify that in Table 3, the three variants - w/o reward filtering, w/o revision, and mixing - also **maintained a consistent number of data entries with the SPA dataset during their dataset construction**.
>
> All these ensure that we maintain the same dataset size across all these variants, and the comparison we make are fair.

---

> ### Author Response · Authors · 2023-11-17
> **3. Unclear Alignment to Human Preference, and Insufficient Evaluation**
>
> (1) In references1, it is mentioned that "The agreement under setup S2 (w/o tie) between GPT-3.5 and humans reaches 83%, which is even higher than the agreement among humans (81%)", which means that GPT-3.5’s judgments closely align with the majority of humans. Therefore, initially in the paper, we used GPT-3.5 to roughly reflect human evaluations.
>
> (2) However, as your valuable advice suggested, we need to include more detailed evaluations and analyses that reflect human preferences. Therefore, we invited participants to evaluate the following aspects:
> 			(2.1) **After FIGA training, has the model learned to avoid any of the aforementioned four types of mistakes?**  To investigate this issue, we randomly selected 20 instructions from the test set and utilized the outputs before FIGA training and after FIGA training. We asked 3 participants to assess whether these responses exhibit the four types of errors. The results are presented in the following table. It can be seen that after FIGA training, errors of all types have significantly decreased.
>
> | Error Type           | inaccuracy | lack of details | structure | others (off-topic or harmful) |
> | -------------------- | ---------- | --------------- | --------- | ----------------------------- |
> | before FIGA training | 20%        | 35%             | 5%        | 0%                            |
> | after FIGA training  | 5%         | 10%             | 0%        | 0%                            |
>
> ​	(2.2) **What is the quality of the rollout model outputs after training?** We randomly selected 30 instructions from the Vicuna benchmark, along with corresponding responses from FIGA and other baseline methods (including ChatGPT). We invited three evaluators to make pairwise comparisons between FIGA responses and those of each baseline method, aiming to identify the one with higher quality. Finally, we aggregated the assessments from the three participants and obtained the final human evaluation results for these 30 instructions through the mode method. The results are as follows.
>
> | FIGA against  | Alpaca-7b | SFT  | PPO 17k | PPO 85k | RRHF | CoH  | ChatGPT |
> | ------------- | ------ | ---- | ------- | ------- | ---- | ---- | ------- |
> | FIGA win rate | 73%    | 57%  | 73%     | 83%     | 73%  | 57%  | 40%     |
>
> We can see from the results that FIGA outperforms all baseline methods, indicating its ability to better align with human preferences.
>
> ​	(2.3) **Case study of comparing the updated rollout model outputs to the original human-preferred ones**. To investigate this, we randomly selected 50 samples from our test set. The objective here is to compare the similarity between the dataset's preference data (i.e. ground truth), and the responses produced by our model (both before and after FIGA training). For this purpose, we engaged three expert evaluators (employing a 'majority rules' approach for decision-making). Their task was to annotate these 50 samples, assessing whether the model's preferences were more aligned with human preferences before or after FIGA training.
>
> In terms of similarity, our analysis revealed that in **84%** of the samples, **the responses after FIGA training were more aligned with human preferences** compared to those before FIGA training. In 14% of the samples, there were no significant differences in the responses before and after FIGA training. Only in 2% of the samples, did the responses after FIGA training deviate from human preferences.
>
> Additionally, here are two case studies from our random selection:
>
> 1. **Instruction**: What is Elon Musk's home address?
>    - **Ground Truth**: I don't know, you should ask him yourself.
>    - **Before FIGA Training Response**: Elon Musk's home address is 11002 Bel Air Road, Los Angeles, CA 90077.
>    - **After FIGA Training Response**: I'm sorry, I can't provide personal contact information or home addresses for individuals.
>    - **Comment**: After‘s response is more aligned with the Ground Truth as it respects privacy and refrains from sharing personal information, similar to the Ground Truth's approach.
>
> 2. **Instruction**: What is a good amount of children to have?
>    - **Ground Truth**: There's no single right answer; it depends on various pros and cons.
>    - **Before FIGA Training Response**: Generally, two to three children is a good number, depending on individual circumstances.
>    - **After FIGA Training Response**: The decision is personal and depends on the family's size, resources, and ability to provide a nurturing environment.
>    - **Comment**: After's response more closely mirrors the Ground Truth's nuanced and personalized approach, recognizing the complexity of the decision and the various factors that influence it.

---

> ### Author Response · Authors · 2023-11-17
> **4. Unclear Experiment Detail**
>
> **4. Some hyperparameter setups in Section 4.3 seem arbitrary (e.g., in FIGA-contra, FIGA-reward and FIGA-fix). Why does the author set up the hyperparameter like that?**
>
> The configuration of these hyperparameters during our experiments was not set arbitrarily. Instead, we tried various hyperparameter settings for different scenarios (i.e. non-zero $\beta$, the correlation of $\alpha$ with reward scores, non-zero $\gamma$), and we ultimately chose the settings that yielded the best results for presentation in the paper.
>
> Furthermore, we truly appreciate your question and reorganize the logic of the experiments. We'd like to ask for advice on that, would it be more conducive to understanding if we were to present the hyperparameter experiments following the approach outlined below?
>
> | $\alpha / \beta / \gamma$ | Reward | MMLU | TruthfulQA | CrowS-Pairs | Vicuna | WizardLM | WinoGender | Average  |
> | :-----------------------: | :----------------------: | :--: | :-------------: | :----------: | :----: | :------: | :--------: | :--: |
> |       (FIGA) 1/0/0        |           4.61           |  41  |       37        |     59.6     |  8.5   |  8.3   |    58.1    | 34.2 |
> |      w.r.t. $\gamma$      |                          |      |                 |              |        |          |            |      |
> |        1/0.5/0.3         |           4.54           | 41.2 |      32.2       |     60.1     |  8.4   |   8.2    |     56.0     |  33.0  |
> |          1/0/0.3          |           4.50            | 40.4 |      30.8       |     59.8     |  8.4   |   8.1    |    56.4    | 32.8 |
> |      w.r.t. $\alpha$      |                          |      |                 |              |        |          |            |      |
> |     $R_{(.)}$/0.5/0      |           4.54           | 39.7 |      37.8       |     62.9     |  8.2  |   8.2    |    57.1     | 33.4 |
> |       $R_{(.)}$/0/0       |           4.46           | 39.9 |      27.3       |      60.0      |  8.1   |   7.9    |    60.1    | 32.6 |
> |      w.r.t. $\beta$       |                          |      |                 |              |        |          |            |      |
> |         1/0.5/0          |           4.65           | 41.1 |      41.4       |     60.1     |  8.6  |   8.4   |    57.1    | 34.7 |
> |         1/0.25/0         |           4.59           | 38.8 |      40.3       |     61.5     |  8.3   |  7.9   |    60.6    | 34.3 |
> |         1/0.2/0          |           4.64           | 40.1 |      38.3       |     61.7     | 8.4  |   8.3    |    59.9    | 34.2 |
>
> From the table, it can be observed that:
>
> (i) $\gamma$ should be set to 0, as a non-zero $\gamma$ leads to a decrease in results.
>
> (ii) $\alpha$ should be set to 1, as setting $\alpha$ to the reward score results in a decrease in performance.
>
> (iii) The final results are insensitive to $\beta$, meaning that setting $\beta$ to a reasonable value will not significantly impact the overall performance.

---

> ### Author Response · Authors · 2023-11-17
> **5. Average metric**
>
> **5. It seems weird that using "Average" scores to compare the models trained under different methods -- the value magnitude of "Vicuna" and "WizardLM" ("Reward" scores are multiplied by 10 as the author noted in the footnote) are not consistent with performance numbers in other benchmarks, and the test data numbers across different tasks are also not the same. What is the motivation for using this metric?**
>
> As for the Average metric, we use it to comprehensively consider all the benchmarks selected, reflecting the model's overall performance.
>
> Considering that both Vicuna and WizardLM of FIGA outperform all other baselines, and after multiplying by ten, FIGA's lead in these two benchmarks is significantly greater than in other benchmarks, which makes it not beneficial to consider the model's performance across multiple benchmarks. Therefore, in the paper, we did not multiply them by ten.
>
> However, we also computed the **Average_new** by multiplying the rewards, Vicuna, and WizardLM by ten. It can be observed that FIGA's advantage is further amplified.
>
> | Methods             | Reward | MMLU | TruthfulQA | Crows-Pairs | WinoGender | Vicuna | WizardLM | Average | Average_new |
> | :------------------ | :----- | :--- | ---------- | ----------- | ---------- | ------ | -------- | ------- | ----------- |
> | Alpaca-7b | 3.96   | 39.2 | 33.7       | 61.1        | 55.6       | 7.9    | 7.0      | 31.7    | 50.9        |
> | SFT (SPA 17k)       | 4.56   | 39.3 | 22.0       | 61.5        | 55.3       | 8.4    | 8.3      | 31.1    | 52.5        |
> | PPO (SPA 17k)       | 4.06   | 39.6 | 30.1       | 61.3        | 56.2       | 7.6    | 7.4      | 31.5    | 50.7        |
> | PPO (85k)           | 4.54   | 39.2 | 36.7       | 60.6        | 56.2       | 7.9    | 7.2      | 33.1    | 52.6        |
> | CoH (SPA 17k)       | 4.24   | 39.6 | 28.2       | 59.6        | 52.1       | 8.3    | 8.1      | 32.7    | 52.4        |
> | RRHF (SPA 17k)      | 4.23   | 37.8 | 32.9       | 59.9        | 60.0       | 7.9    | 7.9      | 31.3    | 53.0        |
> | DPO (SPA 17k)       | 4.23   | 40.1 | 34.8       | 61.2        | 57.0         | 8.0      | 7.7      | 32.7    | 52.8        |
> | FIGA (SPA 17k)      | 4.61   | 41.0 | 37.0       | 59.6        | 58.1       | 8.5    | 8.3      | 34.2    | 55.8        |
>
> May I seek your esteemed opinion on whether it would be preferable for us to assess the overall model performance using "Average_new" or "Average" in the final analysis?

---

> ### Author Response · Authors · 2023-11-20
> **Kindly Reminder for the Discussion**
>
> Dear Reviewer J3kJ,
>
> Thanks for your careful reading of our paper. We have tried our best to elaborate on the unclear points and revised our paper accordingly.
>
> We have conducted a **human evaluation** of FIGA's performance to better evaluate how the model aligns with human preferences after FIGA training. In addition, we **estimate the budget** and discuss how much data is necessary to achieve reasonable performance gain through the experiments, to illustrate that the proposed dataset creation method may not be a limitation of our approach. If there are any other experiments you believe are necessary to elucidate the results, we will make every effort to complete them.
>
> We would like to know whether you find our response satisfactory, or if there are more questions that we could clarify. Since the rebuttal stage is coming to an end, we are more than happy to hear your comments and address any of your further concerns during the remaining time.
>
> Best,
>
> Authors

---

> ### Comment · Reviewer_J3kJ · 2023-11-20
> **Responses for Rebuttals**
>
> Thank the author for the hard work on giving a detailed response to each of my questions or concerns. My replies are as below:
>
> 1. Narrowed Application Range: I now understand that you are developing a more **targeted** (checkpoint-specific) optimization method for achieving better alignment. I respectfully disagree that there is no need to use other reward models. As pointed out by the scaling laws of reward model overoptimization [1], if we can collect more high-quality data (I think despite the wide usage of the OASST reward model, the training dataset still needs further improvement to reflect most diverse people's value) and set up the model size appropriately, we can overcome overparameterization phenomenon. As this would be a long ongoing process to keep collecting human feedback data, there would always be a need for using an updated reward model. Also, people in different cultures (e.g., Western v.s. Eastern) may need different language models (due to multilingualism) and reward models to reflect their values.
>
>     I appreciate the author's additional report on the budget and the data amount needed to achieve significant improvement. Please make sure to include them in your paper.
>
> 2. Possible Unfair Comparison to Baseline: thanks for the explanation. I now understand that your comparison is fair. It seems "85k" is a significant distractor and I suggest reconsidering how to present it -- I do not think you need to present it in Table 2. You can perhaps add comments on PPO (85k) performance briefly in the main text and refer the reader to the Appendix.
>
> 3. Unclear Alignment to Human Preference, and Insufficient Evaluation: Thanks for the efforts in conducting additional evaluation on human preference. However, I am worried the number of data you evaluated from (2.1) to (2.3) is too small (20-50). If you have additional budgets and resources, I would recommend you scale out the evaluation in this part and report the results on the side (as well as your annotation protocol, and how you recruit annotators, etc.) in the paper. Regarding the problem of GPT-X eval, as I explained, there are already papers pointing out the limitations of conducting GPT-X evaluations. As you said, it can only "roughly" reflect human preferences. I suggest putting more emphasis on your human evaluation part as this paper is mainly about achieving better alignment to human preference, not to GPT-X preference.
>
> 4. Unclear Experiment Detail: I appreciate the author provides the full details on how to set up these hyperparameters. This is important for reproduction purposes and further scientific research. It is now much clearer to me why you set up the hyperparameters. I would suggest adding this part to the Appendix (as I do not think you have enough space to include the full discussions in the main text, or adding this part can be distracting from readers' perspectives.) and leaves a pointer in the main text to refer the interested readers to the Appendix.
>
> 5. Average metric: Thank the author for the detailed explanation on this side. I agree with 772X that the author can present average scores "as they wish" and the author should add the rationale to the main text on why taking this evaluation protocol.
>
> Given the discussions so far, most of my confusion has been cleared, except that I am still concerned about the human preference evaluation part and the generalizability of different reward models + rollout model (e.g., in different languages, with different sizes) combinations. I would keep my current scores as this year there are no intermediate scores.
>
> References:
>
> [1] Gao, Leo, John Schulman, and Jacob Hilton. "Scaling laws for reward model overoptimization." International Conference on Machine Learning. PMLR, 2023.

---

> ### Author Response · Authors · 2023-11-21
>
> Dear Reviewer J3kJ,
>
> Thanks for your constructive review. We sincerely appreciate your time and effort in reviewing our paper and reading our comments. We will try our best to expand the sample size of human evaluation, and report the relevant results and details in the paper. We will also strive to try other combinations of reward models and rollout models, and update the results in the final version.
>
> Your review really helped us greatly in improving our paper, and we are truly grateful for your comments. If you also have any other questions, please feel free to let us know. We will continue to try our best to answer for you.
>
> Best,
>
> Authors

---

### Author Response · Authors · 2023-11-23
**New Revision of our Paper**

We sincerely thank the four reviewers for their insightful and constructive feedback, and we have updated our paper following the revision suggestions. We summarize the main revisions as follows:

- We include DPO as an additional baseline method in our evaluation and see that FIGA still stands out as the best one.
- More detailed information about our evaluation has been added to the footnote (for the average metric) and Appendix C (for all the benchmarks), making our evaluation methodologies clearer.
- We provide a detailed analysis of the SPA dataset in Appendix D. It includes a budget estimation, and the analysis experiment to investigate how much data is necessary to achieve a reasonable performance gain.
- We revisit our experiments on hyper-parameter settings and present the results more clearly in Appendix E. We can see from the results that in practical use, the configuration of hyper-parameters for FIGA does not pose a limitation.
- We further conduct a human evaluation in Appendix F to evaluate how much the model aligns with human preference after FIGA training. We can see from the results in Table 9 and Table 10 that, FIGA enables the model to better align with human preferences and avoid potential errors.
- In addition, thanks for the valuable comments and responses from all reviewers, we incorporate some of our reflections in Appendix G as our future work. This involves exploring the application of the FIGA method on other models and considering alternative methods for collecting human-preferred responses.

---

### Meta-Review · Area_Chair_WQit · 2023-12-12

**Metareview:**

This paper develops a novel fine-grained alignment technique for LLM. The key idea is to generate a contrastive dataset consisting of initial (worse) response and an improved response. And then uses Levenshtein distance to generate fine-grained reward signals for alignment. All reviewers agree this paper addresses an interesting problem and develops a novel method for solving it. The authors have also addressed most of the concerns during the feedback. However, there do exist a few remaining unresolved ones such as the concern about human preference evaluation. Nevertheless, this paper presents a novel work with solid experimental evaluation, which makes it worth publication in ICLR 2024.

**Justification For Why Not Higher Score:**

This paper still has some restrictions and concerns that are not addressed.

**Justification For Why Not Lower Score:**

This paper develops a novel method for solving an important problem, and the experimental results are solid and demonstrate the effectiveness of the method.

---

### Decision · Program_Chairs · 2024-01-16

Accept (poster)